# Adaptive laboratory evolution recruits the promiscuity of succinate semialdehyde dehydrogenase to repair different metabolic deficiencies

Hai He [1] ✉, Paul A. Gómez-Coronado [1], Jan Zarzycki [1], Sebastian Barthel [1], Jörg Kahnt[2], Peter Claus[3], Moritz Klein[1], Melanie Klose[1], Valérie de Crécy-Lagard [4,5], Daniel Schindler [6,7], Nicole Paczia [3], Timo Glatter [2] & Tobias J. Erb [1,7] ✉

Promiscuous enzymes often serve as the starting point for the evolution of novel functions. Yet, the extent to which the promiscuity of an individual enzyme can be harnessed several times independently for different purposes during evolution is poorly reported. Here, we present a case study illustrating how NAD(P)⁺-dependent succinate semialdehyde dehydrogenase of *Escherichia coli* (Sad) is independently recruited through various evolutionary mechanisms for distinct metabolic demands, in particular vitamin biosynthesis and central carbon metabolism. Using adaptive laboratory evolution (ALE), we show that Sad can substitute for the roles of erythrose 4-phosphate dehydrogenase in pyridoxal 5'-phosphate (PLP) biosynthesis and glyceraldehyde 3-phosphate dehydrogenase in glycolysis. To recruit Sad for PLP biosynthesis and glycolysis, ALE employs various mechanisms, including active site mutation, copy number amplification, and (de)regulation of gene expression. Our study traces down these different evolutionary trajectories, reports on the surprising active site plasticity of Sad, identifies regulatory links in amino acid metabolism, and highlights the potential of an ordinary enzyme as innovation reservoir for evolution.

Promiscuity or side reactivity describes the inherent feature of enzymes to catalyze more than one reaction and/or accept more than one substrate. Enzyme promiscuity is considered to be a frequent starting point in the evolution of new enzymatic functions and is believed to be essential for the evolution of novel metabolic capabilities[1–3] (Fig. 1a). Because promiscuous activities of enzymes are often rather low, several mechanisms have been proposed by which evolution can repurpose enzymatic side activities under physiological conditions. These include increasing the abundance and/or the catalytic efficiency of an enzyme, so that the promiscuous side activity

[1]Department of Biochemistry and Synthetic Metabolism, Max Planck Institute for Terrestrial Microbiology, Marburg, Germany. [2]Mass Spectrometry and Proteomics Facility, Max Planck Institute for Terrestrial Microbiology, Marburg, Germany. [3]Core Facility for Metabolomics and Small Molecule Mass Spectrometry, Max Planck Institute for Terrestrial Microbiology, Marburg, Germany. [4]Department of Microbiology and Cell Science, University of Florida, Gainesville, FL, USA. [5]Genetic Institute, University of Florida, Gainesville, FL, USA. [6]MaxGENESYS Biofoundry, Max Planck Institute for Terrestrial Microbiology, Marburg, Germany. [7]LOEWE-Center for Synthetic Microbiology, Philipps-University Marburg, Marburg, Germany. ✉e-mail: hai.he@mpi-marburg.mpg.de; toerb@mpi-marburg.mpg.de

**Fig. 1 | Evolutionary trajectories to recruit a promiscuous enzyme for a new function. a** Enzyme promiscuity is a starting point in enzyme innovation. Promiscuous activity of an enzyme $E_p$ on substrate S is often rather low. In certain conditions, such as deactivated primary enzyme E ($\Delta E$), evolution may recruit $E_p$ for the physiological function of E within the evolutionary space shown in (**b**) through various strategies shown in (**c**). **b** Evolutionary space for a promiscuous enzyme ($E_p$) to acquire a new function along a path involving changes in protein abundance ([$E_p$]) and/or catalytic efficiency ($k_{cat}/K_m$). **c** Various mutations in the genome can result in a nonsynonymous substituted enzyme $E_p^*$ with catalytic efficiency, an increase in protein abundance rise from gene amplification or transcription regulation, thus further increasing flux from substrate (S) to product (P) using enzyme $E_p$. TF, transcription factor. Figure elements are created in BioRender[75].

becomes able to carry sufficient flux[1,2,4] (Fig. 1b). To study such evolutionary innovations, adaptive laboratory evolution (ALE) has recently proven a powerful approach[5-7]. While most ALE studies focused on how multiple enzymes can rescue a single metabolic deficiency, the question of to what extent a single enzyme can compensate for different metabolic deficiencies has not been investigated, so far.

The NAD(P)$^+$-dependent succinate semialdehyde dehydrogenase (SSADH) Sad is a member of the ALDH superfamily and oxidizes succinate semialdehyde (SSA) to succinate[8,9]. Some organisms possess (additionally) a strictly NADP$^+$-dependent SSADH[10-12]. In microorganisms, SSADHs operate in the degradation of various nitrogen-containing compounds such as putrescine or arginine, where they prevent the accumulation of the toxic SSA intermediate[8,10,13]. Bacterial SSADHs were also found to participate in lysine catabolism, where it was reported to oxidize glutarate semialdehyde[14,15], and to operate in a non-canonical tricarboxylic acid cycle that proceeds via SSA[16]. Beyond prokaryotes, SSADHs also serve important roles in eukaryotes. In humans, SSADH is involved in the catabolism of a major neuro-transmitter γ-aminobutyrate (GABA) via SSA, and its malfunction causes a metabolic disorder called SSADH deficiency[17]. In plants, SSADH plays important roles in leaf development and morphogenesis[18], as well as in the response to abiotic stress, which results in the formation of toxic intermediates[19].

Here, we show how the intrinsic promiscuity of SSADH, a supposedly highly specialized catabolic enzyme, can be flexibly recruited for different biosynthetic purposes through different mechanisms in evolution (Fig. 1). Employing *Escherichia coli* strains that are auxotrophic to pyridoxine or glycerate, we use ALE to raise different suppressors that we further analyzed through in vitro, in vivo, and omics approaches. Our studies show that all mutations centered around Sad as the functional entity, at the catalytic and regulatory levels (Fig. 1c). Our results show that Sad is a surprisingly flexible enzyme that is not only able to detoxify succinate semialdehyde and glutarate semi-aldehyde but can also functionally substitute erythrose 4-phosphate dehydrogenase in pyridoxal 5′-phosphate (PLP) biosynthesis and gly-ceraldehyde 3-phosphate dehydrogenase (as well as phosphoglycerate kinase) in glycolysis under physiological conditions. Overall, our study illustrates the intrinsic plasticity of an individual enzyme as a reservoir for evolutionary innovation.

## Results
### Pyridoxine auxotrophy can be bypassed by various mutations in ALE
Pyridoxal 5′-phosphate (PLP) is the active form of B6 interconvertible vitamers including pyridoxine (PN), pyridoxal, and pyridoxamine[20,21]. It is an essential cofactor and is used by functionally diverse enzymes,

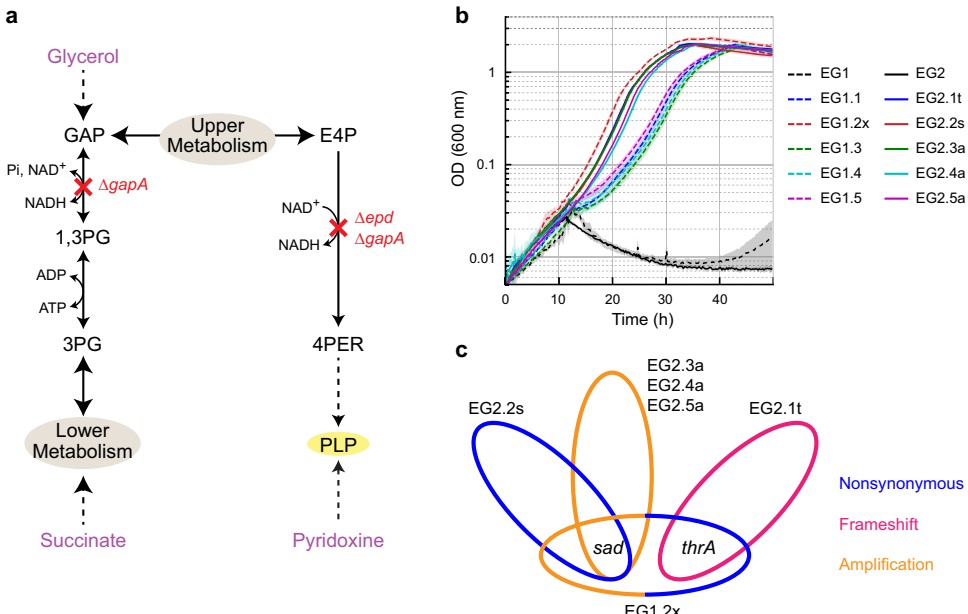

**Fig. 2 | Characterization of evolved Δ*epd* Δ*gapA* strains. a** Schematic representation of the metabolic map of the PN auxotrophic strain, Δ*epd* Δ*gapA* (EG1 and EG2, Supplementary Table 1). Red crosses indicate metabolic steps affected by indicated gene deletions. Carbon sources are indicated in purple. The full metabolic map is shown in Supplementary Fig. 1. Abbreviations, 1,3PG – glycerate 1,3-bisphosphate; 3PG – glycerate 3-phosphate; 4PER – erythronate 4-phosphate; E4P – erythrose 4-phosphate; GAP – glyceraldehyde 3-phosphate; PLP – pyridoxal 5′-phosphate. **b** The parental EG1 and EG2 were not able to grow on minimal medium without PN, while the evolved isolates grew. The details of the strains are listed in Supplementary Table 1. Medium compositions are M9 with 5 mM glycerol and 20 mM succinate. $N = 4$, lines represent mean values and patches indicate SD. **c** Mutations that emerged from adaptive laboratory evolution are only in *sad* and *thrA*, causing protein sequence nonsynonymous change (in blue), frameshift (in pink), or gene amplification (in orange). The detailed mutations are provided in Supplementary Table 2. Source data are provided as a Source Data file.

accounting for ~4% of activities classified by the Enzyme Commission[22,23]. In *E. coli*, genes *epd* and *gapA* encode enzymes that primarily (Epd) or promiscuously (GapA) catalyze the oxidation of erythrose 4-phosphate (E4P), which is the first reaction of the deoxyxylulose 5-phosphate (DXP)-dependent PLP biosynthetic pathway[23,24] (Supplementary Fig. 1). In efforts to construct a PLP auxotroph strain that requires PN supplementation for growth, we created a strain with deletion of these two genes (Δ*epd* Δ*gapA*)[25] in an *E. coli* MG1655 derivative strain SIJ488[26] (Fig. 2a). We picked two independent colonies from the same knock-out step, designated as biological replicates, and labeled as EG1 and EG2 (Supplementary Table 1). While the two strains were not able to grow on M9 minimal medium without supplementation of PN within 40 h, growth of individual replicate cultures could be observed after prolonged incubation (black lines in Fig. 2b, Supplementary Fig. 2a).

Because growth largely varied in replicate cultures, we suspected that it was the result of an adaptive evolution. Given the existence of serendipitous PLP biosynthetic pathways in *E. coli*[27] (Supplementary Fig. 1), we hypothesized that mutations might have activated PLP biosynthetic pathway(s) that would not require E4P oxidation. To elucidate the mutations and the bypassing mechanism(s), we isolated individual suppressor mutants from both EG1 and EG2 (Supplementary Fig. 2b) and additionally conducted an adaptive laboratory evolution (ALE) experiment on solid medium (supplemented with casamino acids, tryptophan, and thiamine) to potentially increase the spectrum of mutations (see "Methods" section and Supplementary Fig. 2c).

We picked a total of 10 suppressor strains (Supplementary Table 1) that all readily grew in the absence of PN, and subjected the six fastest-growing isolates (EG1.2x, EG2.1t, EG2.2 s, EG2.3a, EG2.4a, and EG2.5a; Fig. 2b) to short-read whole genome sequencing. Sequencing reads were mapped to the reference genome of SIJ488 (GenBank: CP132594, see Supplementary Note 1 for de novo assembly). Notably, all mutations identified in the different suppressor strains involved *sad*

and/or *thrA* (Fig. 2c and Supplementary Table 2). EG2.2s had acquired a non-synonymous substitution in *sad*, resulting in an amino acid change of Q262R; while EG2.3a, EG2.4a, and EG2.5a were predicted by the *breseq* pipeline[28] to carry amplified genomic regions that encoded *sad* (-80×, Supplementary Fig. 3a and Supplementary Table 3). An amplification of *sad* was also found in EG1.2x (-30×, Supplementary Fig. 3a and Supplementary Table 3), which additionally possessed a G472S substitution in ThrA. Finally, EG2.1t had accumulated a frameshift in *thrA*, changing downstream amino acid sequence and shortening it by three amino acids (Supplementary Fig. 3b). We next sought to elucidate the biochemical and physiological effects of these individual mutations in more detail.

### Q262R awakens a sleeping E4P dehydrogenase activity in Sad

First, we aimed to elucidate the functional consequences of the mutation(s) of the *sad* gene, which encodes NAD(P)⁺-dependent SSADH. We speculated that Sad was also able to promiscuously oxidize D-erythrose to D-erythronate. We further hypothesized that D-erythrose could be provided upstream from D-erythrose 4-phosphate (E4P) through the known E4P phosphatases YidA and/or YbjI[29], while a serendipitous pathway downstream could convert D-erythronate into the PLP precursor O-phospho-4-hydroxy-L-threonine via known side activities of SerA (phosphoglycerate dehydrogenase), SerC (phosphoserine/phosphorhydroxythreonine aminotransferase), and ThrB (homoserine kinase)[5] (Supplementary Fig. 1).

We next purified the WT and Q262R variants of Sad for biochemical characterization. Both enzymes possessed high activity with SSA (-90 μmol min⁻¹ mg⁻¹), the native substrate, which was unchanged for the Q262R variant compared to the WT (Fig. 3a). We also tested the two enzymes with D-erythrose as substrate (Fig. 3b and Supplementary Fig. 4a). Unexpectedly, the Q262R variant showed a significantly decreased activity ($7.3 \pm 1.2$ nmol min⁻¹ mg⁻¹) with D-erythrose

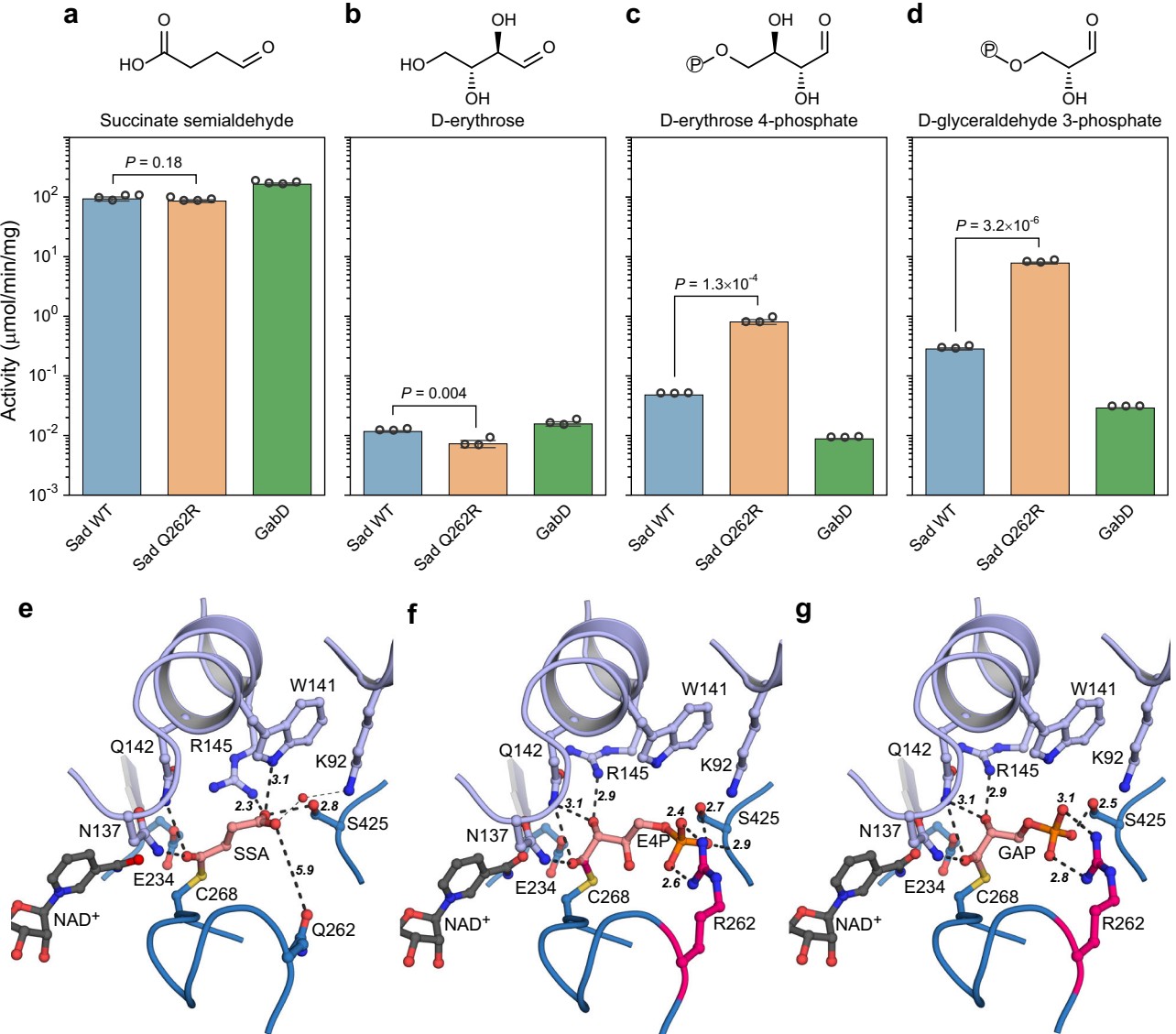

**Fig. 3 | Promiscuous activities of succinate semialdehyde dehydrogenases.** While having high activities on their primary substrate, succinate semialdehyde (SSA, **a**, $N = 4$), NAD$^+$-dependent succinate dehydrogenase Sad and its Q262R mutant as well as NADP$^+$-dependent succinate dehydrogenase GabD showed catalytic activity on D-erythrose (**b**, $N = 3$), D-erythrose 4-phosphate (E4P, **c**, $N = 3$), and D-glyceraldehyde 3-phosphate (GAP, **d**, $N = 3$). The Q262R mutation significantly increased activities on E4P and GAP. Two-tailed $t$-test, $P$-values are shown, and error bars represent SD. **e** The active site of the wild-type Sad with bound NAD$^+$ (dark gray) and substrate (salmon) SSA, the latter is covalently bound to C268. Potential

interactions for the coordination of SSA are shown as black dashed lines and important distances are given in Å. **f, g** The active site of the Q262R variant of Sad with bound NAD$^+$ and modeled E4P & GAP (based on the position of SSA in the Q262R variant (see Supplementary Fig. 6), taking into account available space and potential coordinating residues). Coloring is the same as in (**e**), with the Q262R substitution highlighted in pink. R262 is likely responsible for an improved coordination of the phosphate group of E4P and GAP. Source data are provided as a Source Data file.

compared to the WT ($11.7 \pm 0.4$ nmol min$^{-1}$ mg$^{-1}$), falsifying our initial hypothesis that the Q262R variant showed improved D-erythrose conversion.

Even more surprisingly, when testing the substrate spectrum of Sad, we noticed that the WT enzyme was able to directly accept E4P, notably at four-fold higher activity (50 nmol min$^{-1}$ mg$^{-1}$) compared to D-erythrose. Moreover, in the Q262R variant, E4P dehydrogenase activity was increased 17-fold compared to the WT, reaching 800 nmol min$^{-1}$ mg$^{-1}$ (Fig. 3c). Both variants were also active with glyceraldehyde 3-phosphate (GAP), showing even higher activities with GAP compared to E4P (Fig. 3d). The oxidation of both E4P and GAP was independently confirmed via high-resolution liquid chromatograph-mass spectrometry (see "Methods" section and Supplementary Fig. 4b and c).

To investigate whether this unexpected substrate profile was a more general feature of SSADHs, we also assayed GabD, an alternative, but strictly NADP$^+$-dependent SSADH from *E. coli* (Fig. 3a–d). GabD showed higher activity on SSA than any of the two Sad variants ($164 \pm 9$ μmol min$^{-1}$ mg$^{-1}$) but was almost inactive with E4P and GAP, indicating that the promiscuity with phosphorylated C3 and C4 sugars was restricted to Sad.

Further kinetic investigations revealed that the Q262R substitution had significantly improved the $K_m$ of Sad for sugar phosphates. While for its native substrate, SSA, the Q262R variant showed a two-fold improved apparent (app.) $K_m$, its affinities toward E4P and GAP were increased by 75- and 16-fold, respectively. This improved the catalytic efficiency of the reaction with E4P and GAP by two orders of magnitude (Table 1 and Supplementary Fig. 5), reaching $k_{cat}/K_m$-values

**Table 1 | Kinetic parameters of SSADHs**

| Substrate | Protein | App. $K_m$ (µM) | $k_{cat}$ (s$^{-1}$) | $k_{cat}/K_m$ (s$^{-1}$ M$^{-1}$) | App. $K_i$ (mM) |
|---|---|---|---|---|---|
| Succinate semialdehyde | Sad WT | 11 ± 1 | 85 ± 3 | 7.4 × 10$^6$ | 0.3 ± 0.1 |
| | Sad Q262R | 5 ± 1 | 87 ± 4 | 1.9 × 10$^7$ | 0.5 ± 0.1 |
| | GabD | 13 ± 1 | 132 ± 2 | 1.0 × 10$^7$ | NA |
| Erythrose 4-phosphate | Sad WT | 2,102 ± 328 | 1.1 ± 0.1 | 5.3 × 10$^2$ | NA |
| | Sad Q262R | 28 ± 4 | 3.0 ± 0.1 | 1.1 × 10$^5$ | 7.4 ± 3.3 |
| Glyceraldehyde 3-phosphate | Sad WT | 1,106 ± 92 | 3.1 ± 0.2 | 2.8 × 10$^3$ | NA |
| | Sad Q262R | 67 ± 9 | 19 ± 1 | 2.8 × 10$^5$ | 1.4 ± 0.3 |

Measurements were carried out in 75 mM HEPES pH 8.0, 37 °C in the presence of 1 mM NAD$^+$, or NADP$^+$ in the case of GabD. The data present means ± SD, N = 3, as computed from nonlinear regression (Supplementary Fig. 5). Source data are provided as a Source Data file.
*App.* apparent, *NA* not applicable.

that fall in the range of enzymes on their native substrates (10$^3$–10$^8$ s$^{-1}$ M$^{-1}$)[2,30]. At known intracellular E4P concentration between 15 and 50 µM[31,32], these improved kinetic parameters of the Sad Q262R variant (app. $K_m$ ~30 µM, $k_{cat}$ ~3 s$^{-1}$) would allow for an efficient conversion of E4P at physiological conditions.

To understand the molecular basis of the improved enzyme activity in the Q262R variant, we solved the crystal structures of both Sad WT and Q262R variants with bound NAD$^+$, as well as in complex with both NAD$^+$ and SSA (Supplementary Table 4 and Supplementary Fig. 6). While we were not able to obtain structures with sufficient electron densities for the alternative sugar-phosphate substrates, we manually fitted E4P and GAP based on the NAD$^+$/SSA-bound structures (Fig. 3e–g). These analyses showed that the Q262R substitution likely altered the electrostatic charge in the active site to better accommodate the carboxyl group of SSA, and/or the phosphate group of E4P (Fig. 3f) and GAP (Fig. 3g), explaining the dramatically improved kinetic parameters for the E4P dehydrogenase reaction in Sad.

### Amplification of sad increases Sad levels and also rescues pyridoxine auxotrophy

In strains EG2.3a, EG2.4a, EG2.5a, as well as EG1.2x, *sad* (and surrounding genomic regions) had been predicted to be amplified during ALE (Fig. 2c and Supplementary Table 2). We hypothesized that this increase in gene dose resulted in increased intracellular Sad activity so that Sad could also oxidize sufficient E4P in vivo. This was in line with our biochemical characterization of Sad, which had shown that the WT enzyme already exhibited physiologically relevant kinetic parameters with E4P, in particular an app. $K_m$ ~2 mM that was comparable to that of native Epd (app. $K_m$ 0.5–1 mM; $k_{cat}$ ~200 s$^{-1}$)[24,33].

Yet, to directly provide evidence that the growth of these four strains, which carried *sad* amplification (Fig. 2c and Supplementary Table 2), was due to increased enzyme levels, we analyzed the proteomes of the evolved strains using the Δ*gapA* strain (Supplementary Table 1) as reference. The Δ*gapA* strain is not PN auxotrophic but requires the same carbon sources, glycerol, and succinate for growth. Indeed, all strains had significantly increased levels of Sad (Fig. 4a and Supplementary Fig. 7). In EG2.3a, EG2.4a, and EG2.5a, Sad was overexpressed approximately by 30-fold, while EG1.2x, which grew the best (Fig. 2b), showed a more than 300-fold increased expression of Sad (note that the EG1.2x strain also carried an additional mutation in *thrA*, see below). In contrast, in the EG2.2s strain that harbored the Sad Q262R variant, the Sad expression level was unchanged, supporting the finding that the single amino acid substitution was sufficient to increase E4P dehydrogenase flux (Fig. 4a and Supplementary Fig. 7). Furthermore, both E4P phosphatases, YidA and YbjI, and the alternative SSADH, GabD, showed no significant change in abundance in

any of the strains, suggesting that Sad amplification alone was able to rescue growth by directly substituting the missing E4P dehydrogenase reaction in the knockout strains.

### Reverse genetics confirm that Sad (and GabD) are able to relieve pyridoxine auxotrophy

To further confirm the above hypothesis, we created a pyridoxine auxotrophic (pydxnA) strain by knocking out both SSADH genes, *sad* and *gabD*, on top of Δ*epd* Δ*gapA*. We also knocked out *thrB* to block known serendipitous PLP biosynthetic pathways[5,34] (Fig. 4b and Supplementary Table 1), resulting in an additional threonine auxotrophic phenotype. Upon feeding threonine, glycerol, and succinate, the growth of the strain was PN-dependent. Unlike for EG1 and EG2, no growth was observed in the pydxnA strain even after prolonged incubation (over a week, Supplementary Fig. 8a). The maximal optical densities (OD, at 600 nm) were strictly dependent on PN and correlated closely with PN concentrations down to the nanomolar range. 100 nM of PN was enough to fully restore growth (Fig. 4c,d and Supplementary Fig. 8b). Fitting to the Hill function[35] determined a half maximal effective concentration (EC$_{50}$) of 34 nM for PN (Fig. 4d).

Within this tight and sensitive strain, we tested the complementation through different SSADHs. For this purpose, Sad WT, Sad Q262R, and GabD were expressed from plasmids (Supplementary Table 5) under the control of P$_{LlacO-1}$ promoter[36]. Sad Q262R fully restored growth of the pydxnA strain in the absence of PN even without IPTG induction, while Sad WT and GabD both required induction (Fig. 4e, f), at IPTG levels resulting in ~200-fold increased protein level compared to the non-induced state[37]. Compared to Sad WT, GabD only supported slow growth (Fig. 4e, f), likely due to its lower activity on E4P (Fig. 3c) as well as the rather unfavorable intracellular NADP$^+$/NADPH ratio to operate the oxidation reaction[38]. Reintroduction and overexpression of *thrB* under IPTG induction did not rescue growth in the absence of PN (Supplementary Fig. 9), indicating that the known serendipitous PLP biosynthetic pathways require other components, e.g. high levels of glycolaldehyde or 4-hydroxy-L-threonine[25,34], for sufficient PLP biosynthesis. Overall, the results confirmed that SSADHs, and especially Sad WT and its Q262R variant, can complement Δ*epd* Δ*gapA* and substitute the function of E4P dehydrogenase for PLP biosynthesis.

### Mutations in ThrA indirectly complement pyridoxine auxotrophy by up-regulating sad expression

Next, we sought to investigate strains EG2.1t and EG1.2x that carried mutations in *thrA*, which encodes bifunctional aspartokinase (AK)/homoserine dehydrogenase (HDH) 1 (Fig. 2c and Supplementary Table 2). In both strains, mutations were located within the HDH domain of the protein, resulting in a truncated ThrA in EG2.1t and an amino acid change of glycine472 to serine in EG1.2x (G472S).

We reasoned that these mutations in the HDH domain of ThrA would eliminate HDH activity, and thus decrease production of homoserine. Homoserine is a known inhibitor for ThrB (homoserine kinase) that acts as 4-hydroxy-threonine kinase in serendipitous PLP biosynthetic pathways[39] (Fig. 5a and Supplementary Fig. 1). We speculated that lowering the homoserine pool would increase 4-hydroxy-threonine kinase activity of ThrB, allowing for (increased) PLP biosynthesis through serendipitous PLP biosynthetic pathways (Supplementary Fig. 1). Indeed, truncation of ThrA completely abolished both HDH and AK activity, while the G472S substitution only deactivated HDH activity (Fig. 5b, c). Moreover, intracellular homoserine concentrations in strain EG2.1t and EG1.2x were lower compared to the control strain Δ*gapA* strain (Supplementary Fig. 10a), and homoserine addition to the media decreased the growth of both strains (Supplementary Fig. 11).

While we had observed strongly (i.e., more than 300-fold) increased Sad level in EG1.2x before, (partially) resulting by *sad*

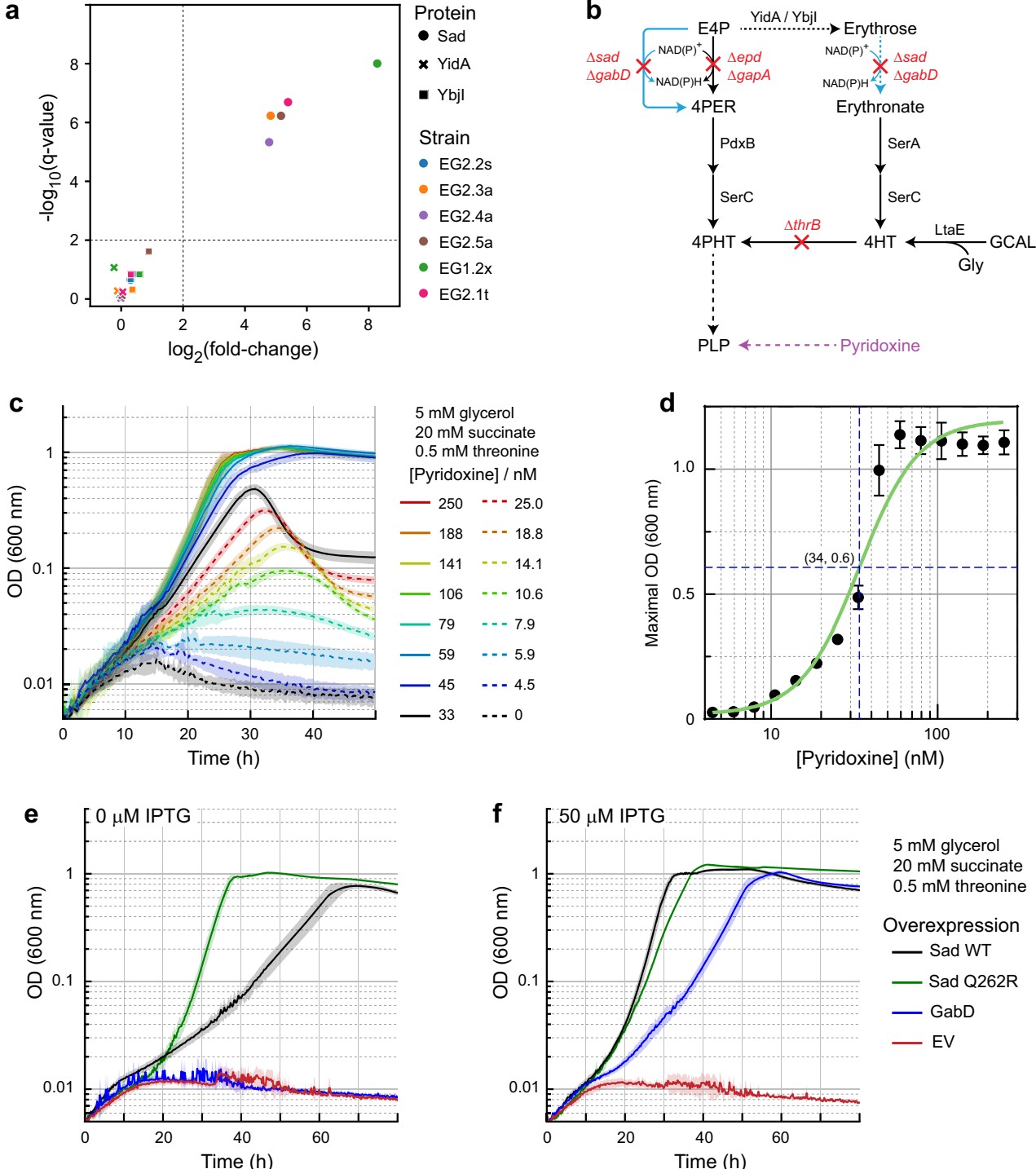

**Fig. 4 | SSADH overexpression complements Δ*epd* Δ*gapA* for growth. a** Scatter plot of log₂ protein intensity levels of Sad and the two E4P phosphatases, YidA as well as YbjI, in the evolved strains relative to the strain Δ*gapA* (X-axis), versus -log₁₀ significance values after *t*-test. Cut-off values highlighting enriched areas (minimal log₂(fold-change) of 2, minimal -log₁₀(*q*-value) of 2) are indicated by the dashed lines. See the "Methods" section for culturing condition and Supplementary Fig. 7 for the whole proteome comparison. **b** Schematic representation of the metabolic map of a tight PN auxotrophic strain. Red crosses indicate metabolic steps affected by indicated gene deletions. The full metabolic map is shown in Supplementary Fig. 1. Abbreviations are the same as Fig. 2, with the addition of 4HT – 4-hydroxy-L-threonine; 4PHT – O-phospho-4-hydroxy-L-threonine; GCALD – glycolaldehyde; Gly – glycine; LtaE – low-specificity L-

threonine aldolase; PdxB – erythronate 4-phosphate dehydrogenase; SerA – phosphoglycerate dehydrogenase; SerC – phosphoserine/phospho-hydro-xythreonine aminotransferase. **c** Growth of such a strain, Δ*epd* Δ*gapA* Δ*sad* Δ*gabD* Δ*thrB* (pydxnA, Supplementary Table 1), is PN concentration dependent. Lines represent mean values and patches show SD, *N* = 6. **d** The response function of the strain to PN. Maximal ODs (*N* = 6, two biological replicates with triplicates each, error bars indicate SD) and PN concentrations were fitted to the Hill function by nonlinear regression using GraphPad Prism (*R²* = 0.9683, see the "Methods" section). EC₅₀ was estimated to be 34 nM. **e, f** Overexpression of Sad, Sad Q262R and GabD restored growth in the absence of PN supplementation. Lines represent mean values, and patches show SD, *N* = 4. EV represents empty vector control. Source data are provided as a Source Data file.

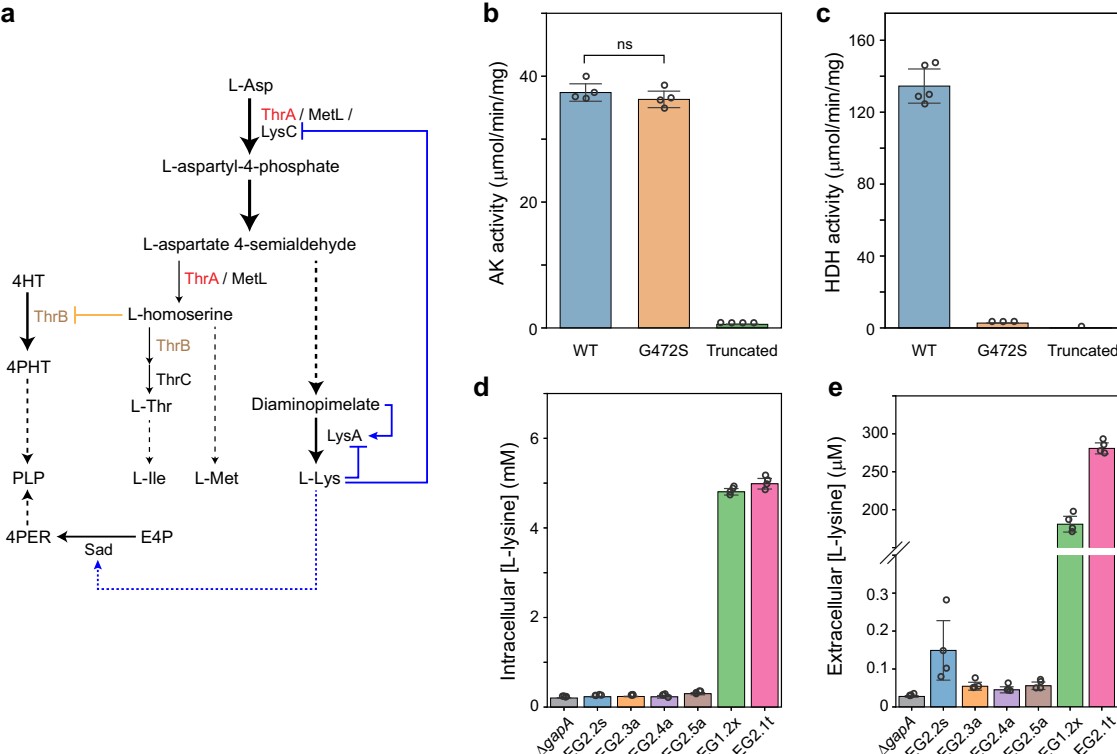

**Fig. 5 | Mutations in *thrA* redistributed metabolic flux to lysine biosynthesis.**
**a** Schematic representation of aspartate-derived amino acid biosynthesis pathways. Regulations at the biochemical level (orange) or the transcriptional level (blue) are shown. Abbreviations of metabolites are the same as Fig. 4, with the addition of amino acids three letter codes. **b** Aspartate kinase (AK) activity of ThrA WT and mutants. *N* = 4, two-tailed *t*-test. **c** Homoserine dehydrogenase (HDH) activity of ThrA WT (*N* = 5) and mutants (*N* = 3). The truncated ThrA had no measurable activity, 0 was used in the plot just for comparison. Intracellular (**d**) and extracellular (**e**) lysine concentrations significantly increased in *thrA* mutant strains, i.e. EG1.2x and EG2.1t (Supplementary Table 1). See also Supplementary Fig. 10b, replicate #2 is shown here, *N* = 4. All error bars represent SD. Source data are provided as a Source Data file.

amplification, we were surprised to learn that in the EG2.1t strain Sad level was also increased by more than 40-fold (Fig. 4a). This suggested that the *thrA* mutations did not induce the known serendipitous PLP biosynthetic pathways, but (indirectly) caused *sad* overexpression, which in turn stimulated PLP synthesis in EG1.2x and EG2.1t. In the case of the EG2.1x strain that also carried a *sad* amplification, we speculated that the ThrA mutation exerted an additional effect on *sad* amplification.

We next aimed to identify the link between ThrA and Sad. Note that homoserine and diaminopimelate (DAP)/lysine biosynthesis branch off from the same intermediate, aspartate 4-semialdehyde (Fig. 5a). We speculated that deactivation of the HDH domain of ThrA would redistribute flux from homoserine towards DAP/lysine biosynthesis. Indeed, in strains EG1.2x and EG2.1t, which had shown decreased homoserine pools (see above), DAP and lysine levels were increased between 2- and 9-fold, respectively (Fig. 5d and Supplementary Fig. 10), and lysine was even secreted into the medium (Fig. 5e). Complementarily, proteomics showed that LysA was significantly up-regulated, while LysC was down-regulated in these two strains, supporting the idea that DAP/lysine metabolism was affected (Supplementary Fig. 7). This was further supported by the fact that externally provided 0.2 mM DAP or lysine both restored growth of the parental strain Δ*epd* Δ*gapA* in the absence of PN (Supplementary Fig. 12), although this phenotype was observed only on glycerate, not succinate, as a carbon source. Interestingly, SSADHs were previously linked to lysine catabolism in *E. coli*: the *csiD* operon, encoding GabD was shown to be induced during lysine degradation and both GabD and Sad were shown to accept glutarate semialdehyde as substrate, a pathway intermediate in lysine catabolism[14]. Altogether, this data suggested that mutations in *thrA* cause reduced homoserine and

increased lysine levels, which in turn up-regulate *sad* expression (Fig. 5a).

## Mutation in YneJ increased Sad level for GAPDH activity
Having shown the surprising flexibility of Sad in rescuing E4P/PLP metabolism, we report in the following on the capability of Sad to substitute another metabolic deficiency, that of GAP dehydrogenase (GAPDH). In an endeavor to establish the tartronyl-CoA (TaCo) pathway[40], a new-to-nature photorespiration module, we created a glycerate auxotroph strain P3P (Supplementary Table 1). This strain was based on MG1655, blocked in glycolysis and gluconeogenesis[41] (Fig. 6a), and carried an additional knockout of the *patZ* gene. *patZ* encodes peptidyl-lysine *N*-acetyltransferase, which we had introduced to suppress post-translational inactivation of one of the TaCo pathway enzymes[40,42]. The P3P strain (Supplementary Table 1) required glycerate for the biosynthesis of serine and phosphoenolpyruvate (PEP) to allow growth (dashed lines in Fig. 6b).

During ALE, we isolated an evolved strain named P3Pe (Supplementary Table 1) that was able to grow independently of glycolate (or glycerate) feeding (solid black line in Fig. 6b). Whole genome sequencing revealed three (most likely irrelevant) mutations (see Supplementary Note 2), as well as one E97A amino acid exchange mutation in the LysR-type transcription factor YneJ (Fig. 6a and Supplementary Table 6). Notably, YneJ has been recently shown to regulate the expression of the *sad* operon (*sad*-*glsB*-*yneG*)[43–45] (see genome context in Supplementary Fig. 3a), which is induced by glutamine[45]. Because E97 is located in the effector binding domain of YneJ[46] (Supplementary Fig. 13a) and LysR-type transcription factors are known to bind diverse effectors, including sugar phosphates and amino acids[47,48], we speculated that the substitution may increase/alter

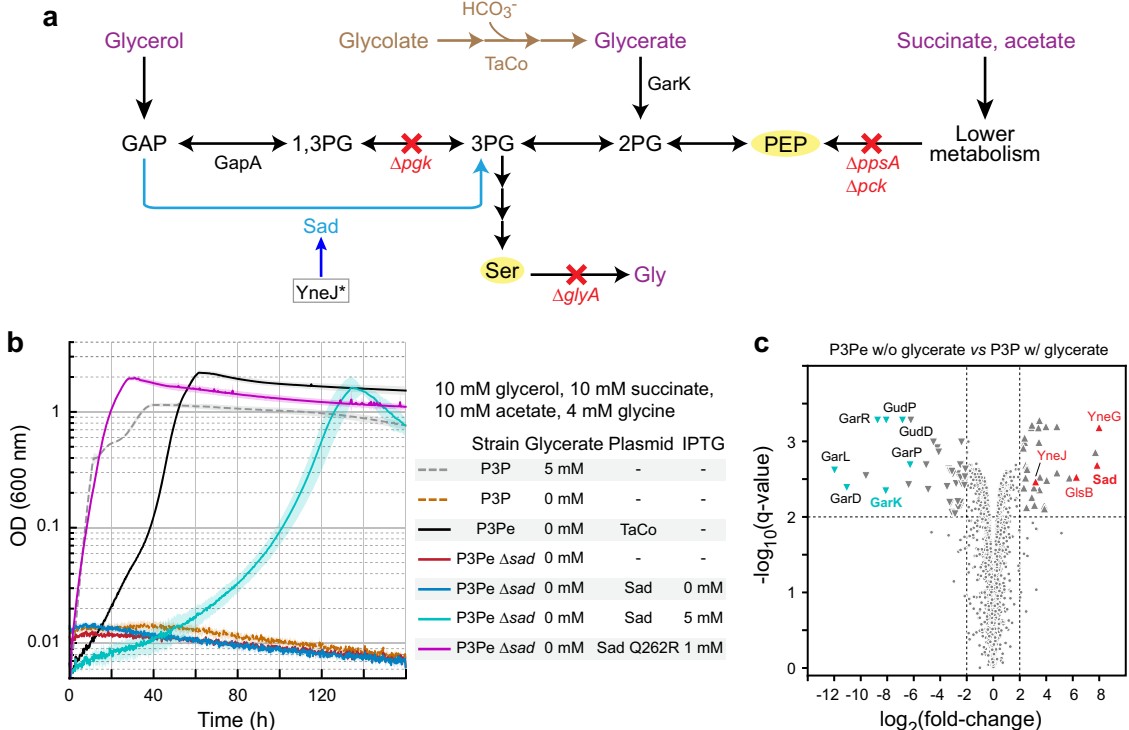

**Fig. 6 | Sad overexpression restored the growth of glycerate sensor strain P3P.**
**a** Schematic representation of the metabolic map of the P3Pe strain. The TaCo pathway[40] is in brown. Red crosses indicate metabolic steps affected by indicated gene deletions. Carbon sources are in purple. Selective metabolites are highlighted in yellow. The E97A mutation in YneJ (framed, Supplementary Table 6) results in high expression of the *sad* gene (blue arrow). Abbreviations are the same as Fig. 2, with the addition of 2PG – 2-phosphoglycerate; GarK – glycerate 2-kinase; Gly – glycine; PEP – phosphoenolpyruvate; Ser – serine. **b** Sad is the responsible enzyme for the growth of the evolved P3Pe strain under glycerate-absent conditions. 50 µM IPTG was used in the preculture. *N* = 4. Lines represent mean values, and patches show SD. **c** Volcano plot representing the comparative proteome of the evolved P3Pe strain relative to its parental P3P strain (Supplementary Table 1). The over-expressed *sad* operon genes are highlighted in red, and the decreased *garK* operon genes are highlighted in cyan. We note that since the parental P3P strain can grow only on medium with glycerate, its expression of *garPLRK*, *garD*, and *gudPD* was up-regulated[76], thus exhibiting decreased expression levels of these genes for P3Pe when it grew on medium without glycerate. *N* = 2. The enrichment areas are high-lighted with value settings as described above. The comparative proteome of P3Pe under glycerate condition is shown in Supplementary Fig. 14. Source data are provided as a Source Data file.

the expression level of *sad*, thus allowing efficient GAPDH activity of Sad to bypass the deletion of *pgk* (Fig. 6a).

Indeed, proteome analysis showed that in P3Pe, the *sad-glsB-yneG* operon seemed to be consistently up-regulated, the levels of all proteins were increased by one and two orders of magnitude compared to the parent strain P3P (Fig. 6c and Supplementary Fig. 14). Knocking out *sad* in P3Pe directly abolished growth (solid red line in Fig. 6b), indicating the role of Sad as GAPDH. Overexpressing *sad* from a plasmid in strain P3Pe Δ*sad* under high concentrations of IPTG rescued growth (light blue line in Fig. 6b and Supplementary Fig. 15). Additionally, overexpressing Sad Q262R instantly enabled growth, with similar behavior to that of the control strain (purple line in Fig. 6b). When further testing for full glycolytic flux, only Sad Q262R, but not the WT Sad or GabD, was able to support growth (solid lines in Supplementary Fig. 16). In summary, this data showed that the E97A substitution in YneJ induced *sad* expression, which substituted GAPDH in P3Pe to overcome the strain's glycolytic deficiency.

## Discussion

Here, we present a case study that illustrates how the promiscuity of a single enzyme can serve as a reservoir for evolution. We show that Sad, which primarily functions in the detoxification of succinate semi-aldehyde in various degradation pathways, does not only accept closely related substrates[49,50], but also converts phosphorylated sugars as a side reactivity. This expands our knowledge of the active site flexibility of SSADHs and highlights the potential of this enzyme to serve in

other completely unrelated metabolic contexts, aligning with the understanding that structural mutations impact genes acting in seemingly unrelated pathways[51].

Although in our experiments we had used an artificial laboratory setup, i.e., metabolic selection strains in combination with ALE, to recruit the sugar-phosphate side activity of SSADHs for new metabolic functions, we note that nature might have already exploited these enzymes naturally during evolution. For instance, GapN, a non-phosphorylating GAPDH that – like SSADH – belongs to the ALDH superfamily[52], and shares 31% identity with Sad, has been apparently recruited to glycolysis in *Thermoproteus tenax* and many other organisms[52–54]. Similarly, SSADHs homologs, and in particular Sad, may fill the gap in PLP biosynthesis in many organisms (e.g., cyanobacteria) that lack Epd, as well as enzymes of the DXP-independent and PLP salvage pathways, but still encode all other DXP-dependent PLP biosynthetic pathway enzymes[55]. This might be the case for different *Microcystis* species, where *sad* is found in the same operon as *pdxA*, which encodes 4-hydroxythreonine 4-phosphate dehydrogenase in the DXP-dependent pathway, indicating that the organism might have recruited Sad for PLP biosynthesis during evolution (Supplementary Fig. 17).

Equally surprising as the intrinsic promiscuity of Sad was the discovery that ALE used various mechanisms to recruit the enzyme to PLP biosynthesis or glycolysis, respectively. Besides acting on the enzyme itself (e.g., active site mutation Q262R, gene amplification), evolution also harnessed several indirect mechanisms (i.e.,

regulatory mutations), revealing regulatory links in amino acid metabolism (e.g., the influence of homoserine and lysine levels on Sad expression), in agreement with that regulatory mutations, affect genes acting in pathways relevant to the novel function[51]. This highlights not only the strong interdependence of the metabolic and regulatory layers of biological networks but also the capability of these intertwined systems to react to new challenges. In respect to these observations, one might speculate that enzymes that sit at the crossroads of different metabolic and/or regulatory nodes and that are regulated through multiple mechanisms might be more rapidly recruited by evolution, so that not necessarily the best, but the most likely candidate would be repurposed during evolution[6,56–58]. Similarly, one might also carefully speculate that all-rounder enzymes, such as SSADH, which serve in the (background) metabolism of reactive metabolites and possess some intrinsic promiscuity to react on various structurally related compounds, could potentially be predestined as innovation hotspots that are prone to adopt new metabolic functions. Yet, such speculations would need to be further tested to understand the inherent adaptability and versatility of biological systems, and in particular the role of catalytic promiscuity as a reservoir for enzyme innovation.

## Methods

### Chemicals

D-(-)erythrose (70% solution, TCI, Germany) and D-erythrose 4-phosphate (E4P) (Sigma-Aldrich, Germany) were prepared as 40 mM stocks and de-dimerized at 4 °C overnight before storing at −20 °C, as they form dimers at high concentrations[59,60]. The concentration of E4P was titrated by assaying with *E. coli* D-erythrose-4-phosphate dehydrogenase (Epd). 25 mM L-aspartic 4-semialdehyde was obtained from Biosynth (Switzerland), and 15% succinic semialdehyde solution from Santa Cruz. ATP (adenosine 5′-triphosphate disodium salt hydrate), L-homoserine, and 20% D/L-glyceric acid were from TCI, the latter was neutralized with NaOH when preparing stock solution. 2,6-diamino-pemelic acid, $NAD^+$ (β-nicotinamide-adenine dinucleotide hydrate), PEP (phosphoenolpyruvic acid monopotassium salt), and NADH (β-nicotinamide-adenine dinucleotide, reduced disodium salt hydrate) were from Sigma-Aldrich. $NADP^+$ (nicotinamide-adenine dinucleotide phosphate monosodium salt) and NADPH (β-nicotinamide-adenine dinucleotide 2′-phosphate, reduced, tetrasodium salt) were from PanReac AppliChem (Germany). Other chemicals were all from Sigma-Aldrich if not specified.

### Growth media

LB medium (Roth, Germany) was used in cultivating cells for gene knocking out and plasmid construction. TB medium (12 g/L tryptone, 24 g/L yeast extract, 4% glycerol, 2.31 g/L $KH_2PO_4$, and 12.54 g/L $K_2HPO_4$) was used for protein production. M9 minimal medium (Sigma-Aldrich, Germany) supplemented with 2 mM $MgSO_4$, 100 μM $CaCl_2$, and trace elements (134 μM EDTA, 31 μM $FeCl_3$, 6.2 μM $ZnCl_2$, 0.76 μM $CuCl_2$, 0.42 μM $CoCl_2$, 1.62 μM $H_3BO_3$, 0.081 μM $MnCl_2$) was used for growth experiments. Carbon sources and supplements used were indicated in the main text. Strains with ΔgapA were maintained on Medium X, which is M9 supplemented with 5 g/L casamino acids, 4 mM glycerol, 40 mM succinate, 1 μg/mL thiamine and 20 μg/mL tryptophan. 1 μM pyridoxine was further supplemented for pyridoxine auxotrophic strains. Strains with Δpgk were maintained on Medium Y, which is M9 supplemented with 5 g/L casamino acids, 10 mM glycerol, 10 mM succinate, and 10 mM acetate. For M9 and Medium X/Y solid media, 1.2% Ultra-pure agarose (Invitrogen, USA) was used. If needed, antibiotics were used at concentrations of 50 μg/mL kanamycin, 100 μg/mL ampicillin, 100 μg/mL streptomycin, 20 μg/mL gentamycin, or 30 μg/mL chloramphenicol in rich media, and half the concentrations in M9 minimal medium. Liquid Hi-Def Azure medium was from Teknova (US).

### Strains and genomic manipulation

All strains used in the study are listed in Supplementary Table 1. Gene knockouts were obtained by λ-Red recombination or P1 phage transduction[61]. Recombineering knock-out cassettes were generated by PCR amplification of pKD4 (GenBank: AY048743) and pKD3 (Gen-Bank: AY048742) plasmids for kanamycin resistance (Km) and chloramphenicol resistance (CAP) cassettes[62], respectively, using 50 bp homologous arms from the target genes in the primers (Supplementary Data 1). Approximately 300 ng of cassette DNA was transformed into freshly prepared electrocompetent cells, which were induced 45 min prior with 15 mM L-arabinose when $OD_{600}$ was around 0.3. The knock-out colony was selected on kanamycin or chloramphenicol plates and verified by colony PCR with "Ver" primers (Supplementary Data 1). Flippase was induced by 50 mM L-rhamnose to remove antibiotic markers. And the removal was confirmed by colony PCR[61].

Strains with ΔgapA and P3 as well as its derivative strains were selected and maintained on Medium X or Medium Y, respectively. λ-Red recombination constructed ΔgapA::CAP was used in P1 transduction for obtaining other strains (Supplementary Table 1). Liquid Hi-Def Azure medium with 5 mM glycerol and 20 mM succinate was used in the processes. P3 derivative strains were constructed by P1 transduction and antibiotic resistance marker removal was assisted with pSIJ8 plasmid.

### Growth experiments

Growth experiments started with preculturing the cells in 3 mL M9 medium in glass tubes under relaxing conditions. Overnight dense cultures were short time kept at 4 °C and washed five times with M9 salt solution by centrifugation at 8000 × g, 10 °C for 3 min. Cells were inoculated in dedicated testing conditions in 96-well plates (Nunclon Delta Surface, Thermo Scientific) at starting $OD_{600}$ of 0.005 cultured at 37 °C in a microplate reader (BioTek Synergy H1). Each well contains 150 μL culture and is covered with 50 μL sterilized mineral oil (Sigma-Aldrich, Germany). The shaking program cycle (controlled by Gen5 v3.11) has 4 shaking phases, lasting 60 s each: linear shaking followed by orbital shaking, both at an amplitude of 3 mm, then linear shaking followed by orbital shaking both at an amplitude of 2 mm. The optical density ($OD_{600}$) in each well was monitored and recorded after every three shaking cycles (-13.5 min). Raw data from the plate reader were calibrated to normal cuvette measured $OD_{600}$ values according to Eq. 1:

$$OD_{cuvette} = OD_{plate}/0.25 \qquad (1)$$

Python script (https://github.com/he-hai/growth2fig) was used for calculation and plotting based on at least three technical replicates.

GraphPad Prism v9 was used in data fitting of the pydxnA strain's growth parameters (Supplementary Table 1 and Fig. 4c). The nonlinear fitting method of "Dose-response - Stimulation, variable slope (four parameters)" was used for pyridoxine concentrations – maximal OD, weighted by $1/Y^2$. Pyridoxine concentrations – growth rates (μ) were fitted in the Michaelis–Menten equation without weighting and excluding data when [pyridoxine] is below 15 nM.

### Adaptive laboratory evolution of Δepd ΔgapA strains

In growth experiments, strains EG1 and EG2 (Supplementary Table 1) emerged to grow after around 50 h under conditions where pyridoxine was absent or at a 10 nM level. Liquid culture was streaked onto Medium X plates without pyridoxine to isolate single colonies, resulting in isolates EG1.1, EG1.2x, EG2.1t, and EG2.2s. Additionally, strains EG1 and EG2 (Supplementary Table 1) were cultured on Medium X solid plates without pyridoxine supplementation for adaptive evolution. Cell materials were scraped with inoculation loops when fully grown (around 3 days) and directly re-streaked to fresh Medium X

plates. This procedure was done three times (Supplementary Fig. 2). Three colonies of each strain were isolated for further study.

## Whole genome sequencing and analysis

NucleoSpin Microbial DNA kit (MACHERY-NAGEL, Düren, Germany) was used for genomic DNA extraction following the manufacturer's instructions. Library construction used Nextera XT kit (Illumina) and genome sequencing was on a paired-end Illumina sequencing platform MiSeq. Raw reads have been deposited at the NCBI Sequence Read Archive (SRA) and can be accessed under BioProject PRJNA1010158. The sequencing data were first passed through quality trimming using Trim Galore (v0.6.7)[63], then mapped to a SIJ488 (GenBank: CP132594) or MG1655 (GenBank: NC_000913) as reference genome using *breseq* (v0.36.1)[28,64].

## Plasmid constructions

Plasmids used in the study were listed in Supplementary Table 5. NEB HiFi assembly kit (New England Biolabs, Germany) was used for plasmid constructions. pET(16b) backbone was linearized by BamHI and NdeI (FastDigest, Thermo Scientific). pET(23b+) backbone was linearized by XhoI and NdeI (FastDigest, Thermo Scientific). The pLac backbone and inserts were amplified with PrimeStar GXL DNA polymerase (Takara) using primers in the Supplementary Data 1. Assembly products were transformed to DH5α or Pir1 (Supplementary Table 1) chemical competent cells and selected on appropriate LB selection plates. All plasmids were prepared with a NucleoSpin plasmid kit (MACHERY-NAGEL, Düren, Germany) and confirmed by Sanger sequencing (Microsynth Seqlab GmbH, Germany).

## Recombinant protein production and purification

Recombinant protein production and purification were conducted following generic procedures, with specific changes for each individual protein as detailed in Supplementary Table 7. In general, plasmids (Supplementary Table 5) were transformed to BL21(DE3) (Supplementary Table 1) chemical competent cells and selected on LB plates with the appropriate antibiotic. Colonies from the selection plates were washed down and inoculated to 1 L TB with antibiotic in 5 L baffled flasks. The cultures were grown at 37 °C, 110 rpm to $OD_{600}$-1, then cooled to 25 °C and induced with IPTG. After around 16 h, the cultures were divided into 500 mL portions, pelleted by centrifugation ($8000 \times g$, 10 min), then directly followed by lysing and purification or stored at −20 °C. Cells were resuspended in 3 times (v/w) lysis buffer and lysed by sonication with SONOPULS HD 4000 with a KE 76 probe (BANDELIN electronic GmbH & Co. KG, Berlin, Germany) at 60% amplitude for 5 times 1 min of 1 s on/off pulses. Cell lysates were clarified by centrifugation ($50,000 \times g$, 10 °C for 1 h) and then filtrated through 0.45 µm syringe filters (PES membrane; Sarstedt, Nümbrecht, Germany).

Affinity purification and desalting were performed on an Äkta Start FPLC system (Cytiva, Freiburg, Germany) or manually. Using the FPLC system at 4 °C, filtrate lysate was added 2 M imidazole to the final 25 mM and then loaded onto 1 mL HisTrap FF column (Cytiva, Freiburg, Germany) equilibrated with 95% buffer A, 5% buffer B. The column was then washed with 80% buffer A, and 20% buffer B. Proteins were eluted with 100% buffer B in 1 mL fractions. The protein fractions, judged by $A_{280}$, were pooled and buffer exchange to buffer D was performed on two stacked 5 mL HiTrap desalting columns (Cytiva, Freiburg, Germany).

In gravity manual purification, filtrate lysate was added to buffer A equilibrated 2 mL Ni-NTA agarose (Protino, MACHERY-NAGEL, Düren, Germany), and added imidazole to the final 25 mM. The lysate-beads resuspension was incubated at 4 °C for 30 min with 30 rpm shaking. The resuspension was then flowed through a 14 mL Protino column (MACHERY-NAGEL, Düren, Germany) and agarose beads were washed with 80% buffer A, and 20% buffer B. Proteins were eluted with 100% buffer B in 1 mL fractions. The protein fractions, judged by $A_{280}$ measured on NanoDrop (Thermo Scientific, Waltham, MA, USA), were pooled and buffer exchange to buffer D was performed on PD-10 desalting column (Cytiva, Freiburg, Germany).

Purified proteins were concentrated by ultrafiltration (Amicon, Millipore) at $4000 \times g$, 4 °C. Glycerol was then added to the final 20% (v/v). Protein concentrations were determined via Bradford assay (Quick Start Bradford 1× Dye, BioRad) using BSA as standard. Protein purities were confirmed by SDS-PAGE (Supplementary Fig. 18) using precast 4–20% gels (Mini-PROTEAN TGX, BioRad, USA). The proteins were aliquoted and stored at −20 °C.

## Enzymatic assays

Enzymatic assays were conducted in a 200 µL reaction mixture in quartz cuvettes (1 cm length, Hellama Analytics) at 37 °C. NAD(P)H ($\varepsilon_{340} = 6220$ M$^{-1}$ cm$^{-1}$) production or consumption was monitored by tracking the absorbance at 340 nm using Cary 60 UV-Vis Spectrophotometer (v2.00, Agilent) and Cary WinUV Kinetics Application (v5.0.0.999, Agilent). Unless otherwise mentioned, components were added to the cuvette in the order that they were listed, and the last indicated component was used to initiate the reaction. Initial velocity slopes were obtained by linear regression fitting. For kinetic measurements, data were fit to Michaelis–Menten or substrate inhibition equation using homemade Python scripts (https://github.com/he-hai/enzymatic-tools).

## SSADH activity assay

Sad WT and Sad Q262R activities on different substrates were assayed in 75 mM HEPES-KOH, pH 8.0, 1 mM NAD$^+$, 1 µM enzyme (0.01 µM in the case of the primary substrate succinate semialdehyde), 0.1 mM substrate, i.e. succinate semialdehyde, erythrose, E4P, or glyceraldehyde 3-phosphate (GAP). NADP$^+$ was used in the case of GabD. 7.5 µL of formic acid was added to a 150 µL reaction mixture of phospho-sugars after 10 min or 30 min in the case of erythrose. Removing proteins by centrifugation at $20,000 \times g$, 2 °C for 15 min, the supernatants were subjected to LC-MS analysis.

Kinetics of Sad WT, Sad Q262R, and GabD were determined in the conditions mentioned above, except that the protein concentrations were adjusted to 0.005 µM when on succinate semialdehyde; 0.2 µM and 1 µM for Sad WT on GAP and E4P, respectively; 0.05 µM and 0.1 µM for Sad Q262R on GAP and E4P, respectively.

## Aspartate kinase

Aspartate kinase activities of ThrA variants were determined in 75 mM HEPES-KOH, pH 8.0, 100 mM KCl, 5 mM MgCl$_2$, 1.5 mM PEP, 0.3 mM NADH, 5 U/mL rabbit muscle pyruvate kinase/lactic dehydrogenase (Sigma-Aldrich), 4 mM ATP, 5 mM L-aspartate, and ThrA WT, ThrA G472S at 0.01 µM or ThrA truncated at 1.21 µM.

## Homoserine dehydrogenase

Homoserine dehydrogenase activities of ThrA variants were determined in 75 mM HEPES-KOH, pH 8.0, 100 mM KCl, 5 mM MgCl$_2$, 0.2 mM NADPH, 1 mM L-aspartic 4-semialdehyde, and WT, G472S, or truncated ThrA at 0.002 µM, 0.2 µM, or 1.21 µM, respectively. The activity of truncated ThrA was set to 0 as it was below the instrument sensitivity limit and not measurable.

## LC-MS of enzymatic reaction products

Determination of enzymatic products erythronate 4-phosphate and glycerate 3-phosphate was performed using high-resolution liquid chromatograph-mass spectrometer (HRES LC-MS, Agilent Infinity II 1260 HPLC system). Determination of erythronate was performed using a LC-MS/MS (Agilent Infinity II 1290 HPLC system). The chromatographic separation was performed on HPLC systems using a SeQuant ZIC-pHILIC column (150 × 2.1 mm, 5 µm particle size, peek

coated, Merck) connected to a guard column of similar specificity (20 × 2.1 mm, 5 μm particle size, Phenomenex) a constant flow rate of 0.1 mL/min with mobile phase A with mobile phase comprised of 10 mM ammonium acetate in water, pH 9, supplemented with medronic acid to a final concentration of 5 μM (A) and 10 mM ammonium acetate in 90:10 acetonitrile to water, pH 9, supplemented with medronic acid to a final concentration of 5 μM (B) at 25 °C.

For erythronate 4-phosphate and glycerate 3-phosphate, the injection volume was 3 μL. The mobile phase profile consisted of the following steps and linear gradients: 0–0.5 min constant at 65% B; 0.5–8.5 min from 65 to 20% B; 8.5–11 min constant at 20% B; 11–11.1 min from 20 to 65% B; 11.1–20 min constant at 65% B. An Agilent 6550 ion funnel Q-TOF mass spectrometer was used in negative mode with a dual jet stream electrospray ionization source and the following conditions: ESI spray voltage 2500 V, nozzle voltage 500 V, sheath gas 350 °C at 12 L/min, nebulizer pressure 20 psig and drying gas 150 °C at 11 L/min. Compounds were identified based on their accurate mass (calculated with https://www.chemcalc.org/) within a mass window of 20 ppm. Extracted ion Chromatograms of the [M-H]$^-$ ions were integrated using MassHunter software (v10.0, Agilent, Santa Clara, CA, USA) applying a mass extraction window given above. Relative abundance was determined based on the peak area.

For erythronate, the injection volume was 1 μL. The mobile phase profile consisted of the following steps and linear gradients: 0–1 min constant at 75% B; 1–6 min from 75 to 40% B; 6–9 min constant at 40% B; 9–9.1 min from 40 to 75% B; 9.1–20 min constant at 75% B. An Agilent 6495 ion funnel mass spectrometer was used in negative mode with an electrospray ionization source and the following conditions: ESI spray voltage 3000 V, nozzle voltage 1000 V, sheath gas 300 °C at 11 L/min, nebulizer pressure 20 psig and drying gas 100 °C at 11 L/min. Compounds were identified based on their mass transition and retention time compared to standards. Chromatograms were integrated using MassHunter software (v10.0, Agilent, Santa Clara, CA, USA). Mass transitions, collision energies, Cell accelerator voltages, and Dwell times have been optimized using chemically pure standards. Parameter settings of all targets are given in Supplementary Table 8.

## Crystallization

Sad WT and Sad Q262R were purified using the Äkta-HisTrap system as described in the "Recombinant protein production and purification" section. Immediately after affinity purification, the eluate was loaded onto a HiLoad 16/600 Superdex 200 pg column (Cytiva, Freiburg, Germany) equilibrated in buffer G (20 mM HEPES-KOH, 50 mM KCl, pH 7.5), on an Äkta pure FPLC system (Cytiva, Freiburg, Germany). Fractions corresponding to dimeric protein were collected, pooled, and concentrated using 50,000 MWCO Amicon Ultra filters (Amicon, Millipore) by centrifugation at 4000 × g, 4 °C.

Crystals were grown using the sitting drop vapor diffusion method at 22 °C. For the WT with NAD$^+$ bound dataset: Sad WT (33.6 mg/mL) in 20 mM HEPES-KOH, 50 mM KCl, pH 7.5 was complemented with 5 mM NAD$^+$. The enzyme was then mixed in a 2:1 ratio with 285 mM Bis-Tris-Propane, pH 8.25, 17% (w/v) PEG4000. The final drop size was 3 μL. The drop was complemented with PEG200 to a final concentration of 20% (v/v) before flash-freezing crystals in liquid nitrogen.

For the WT with NAD$^+$ and SSA-bound dataset: Sad WT (7.3 mg/mL) in 20 mM HEPES-KOH, 50 mM KCl, pH 7.5 was pre-mixed with NAD$^+$ (final concentration 5 mM). The pre-mixed protein was then mixed in a 2:1 ratio with 250 mM Bis-Tris-Propane, pH 8.25, 20% (w/v) PEG4000. The final drop size was 3 μL. The crystal was soaked with 5 mM succinic semialdehyde for 2 min and the drop was complemented with PEG200 to a final concentration of 20% (v/v) before flash-freezing crystals in liquid nitrogen.

For the Q262R mutant with NAD$^+$ bound dataset: Sad Q262R (35.0 mg/mL) in 20 mM HEPES-KOH, 50 mM KCl, pH 7.5 was supplemented with 5 mM NAD$^+$ and 1 mM TCEP. The enzyme was then mixed in a 1:1 ratio with 285 mM Bis-Tris-Propane, pH 8.25, 17% (w/v) PEG4000. The final drop size was 4 μL. The drop was complemented with PEG200 to a final concentration of 16% (v/v) before flash-freezing crystals in liquid nitrogen.

For the Q262R mutant with NAD$^+$ and SSA-bound dataset: Sad Q262R (14.8 mg/mL) in 20 mM HEPES-KOH, 50 mM KCl, pH 7.5 was complemented with 5 mM NAD$^+$. The enzyme was then mixed in a 2:1 ratio with 250 mM Bis-Tris-Propane, pH 7.5, 20% (w/v) PEG4000. The final drop size was 3 μL. The drop was complemented with 6 mM succinic semialdehyde and 25% (v/v) PEG400 before flash-freezing crystals in liquid nitrogen.

X-ray diffraction data (Supplementary Table 4) were collected at the beamlines PETRA III P13 & P14 of the DESY (Deutsches Elektronen-Synchrotron, Hamburg), as well as at ID30A-3 (MASSIF-3) of the ESRF (European Synchrotron Radiation Facility, Grenoble). Data were processed with XDS[65]. Structures were solved by molecular replacement using Phaser of the Phenix software package[66], using the YneI from *Salmonella typhimurium* LT2 (PDB 3EFV)[67] as a search model, and refined with Phenix.Refine. Additional modeling, manual refinement, and ligand fitting were done in Coot[68]. Final positional and B-factor refinements, as well as water picking, were performed using Phenix.Refine. Structural models were deposited to the Protein Data Bank in Europe (PDBe) under PDB accessions 8QMQ, 8QMR, 8QMS, and 8QMT. Figures were made using PyMOL (v2.5, Schrödinger, LLC. (www.pymol.org)).

## Metabolomics

The culture was set up as described in the "Whole-cell proteomics" section. At OD$_{600}$ ~ 0.6 (late exponential phase), the culture was spun down at 14,000 × g for 1 min, the supernatants were stored at −20 °C until further exometabolome analysis via LC-MS. For the analysis of intracellular metabolites, 1 mL culture was added to 1 mL −70 °C cooled 70% (v/v) methanol to quench metabolism, followed by centrifugation at 13,000 × g, −10 °C for 10 min. The supernatants were removed with syringes and cell pellets were stored at −70 °C until extraction.

To extract the intracellular metabolome, 0.2 mL cold extraction fluid (50% methanol in 10 mM Trizma, 1 mM EDTA, pH 7.0) and 0.2 mL cold chloroform were added to cell pellets, followed by short vortexing and 2 h incubation at 0 °C with 1000 rpm shaking. After centrifugation at 20,000 × g at −10 °C for 10 min, the aqueous upper phase was taken and stored at −70 °C until LC-MS analysis.

Quantitative determination of diaminopemelate (DAP), lysine (LYS), and homoserine (HSER) were performed by using HPLC-MS/MS. The chromatographic separation was performed on Infinity II 1290 HPLC system (Agilent) using a ZicHILIC SeQuant column (150 × 2.1 mm, 3.5 μm particle size, 100 Å pore size) connected to a ZicHILIC guard column (20 × 2.1 mm, 5 μm particle size) (Merck KgAA), with a constant flow rate of 0.3 mL/min, with mobile phase A being 0.1% formic acid in 99/1 (v/v) water:acetonitrile and phase B being 0.1% formic acid in 1/99 (v/v) water:acetonitrile at 25 °C. The injection volume was 1 μL. The mobile phase profile consisted of the following steps and linear gradients: 0–8 min from 80 to 60% B; 8–10 min from 60 to 10% B; 10–12 min constant at 10% B; 12–12.1 min from 10 to 80% B; 12.1 to 14 min constant at 80% B. An Agilent Triple Quad 6495 ion funnel mass spectrometer was used in positive mode with an electrospray ionization source and the following conditions: ESI spray voltage 2000 V, nozzle voltage 1000 V, sheath gas 250 °C at 12 L/min, nebulizer pressure 60 psig, and drying gas 100 °C at 11 L/min. Compounds were detected using multi-reaction monitoring and identified based on their mass transition and retention time compared to standards. Chromatograms were integrated by using MassHunter software (v10.0, Agilent, Santa Clara, CA, USA), and the Agile2 integrator. Absolute concentrations were calculated based on an external calibration curve

prepared in the sample matrix. The mass transitions, collision energies, fragmentor, and accelerator voltages are reported in Supplementary Table 8.

The calculation of intracellular concentration was according to the Eq. 2:

$$c = \frac{c_m \times V_e}{V_c \times \text{OD} \times E \times V_i} \qquad (2)$$

where $c_m$ is the measured concentration of the extract; $V_e$ is the volume of extraction fluid, that is 0.2 mL; $V_c$ is the volume of harvested culture, that is 1 mL; $OD$ is the optical cell density at 600 nm when harvesting; $E$ is the OD-specific cell concentration ($1.67 \times 10^9$ cells/mL/OD$_{600}$, the value from ref. 69. on M9 succinate) and $V_i$ is the intracellular aqueous volume of an *E. coli* ($6.7 \times 10^{-10}$ μL/cell, value from BioNumbers[70] #108815 and #100011).

### Adaptive laboratory evolution of P3P strain

For the adaptive laboratory evolution of the P3P strain carrying two expression vectors containing the genes for the TaCo pathway, the strain was grown in triplicates 4 mL of M9 media with 10 mM glycerol, 10 mM succinate, 10 mM acetate, 4 mM glycine, 50 mM glycolate, streptomycin, gentamicin and 1 mM glycerate in 15 mL glass tubes. After cells reached a maximum OD-0.15, they were reinoculated in fresh media, and the process was repeated 10 times over the course of 6 weeks. One tube of the triplicates reached an OD-1 after the sixth iteration and was then reinoculated in M9 media with 10 mM glycerol, 10 mM succinate, 10 mM acetate, and 4 mM glycine as negative control. After confirming growth in negative control media, the evolved strain's DNA was isolated for sequencing. The remaining replicates did not reach a higher OD at the end of the 10th iteration when the evolution was stopped.

### Whole-cell proteomics

The evolved strains of EG1 and EG2 as well as Δ*gapA* strain (Supplementary Table 1) were precultured in 3 mL M9 5 mM glycerol, 20 mM succinate, 1 μM pyridoxine, 15 μg/mL chloramphenicol. Overnight dense cultures were washed five times and inoculated into M9 5 mM glycerol and 20 mM succinate, 4 mL in glass tubes, at starting OD$_{600}$ of 0.05, four cultures of each. Cultivated at 37 °C, 180 rpm, 1.5 OD$_{600}$ of cells were harvested at OD$_{600}$-0.6 (late exponential phase) by centrifugation at 20,000 × g, 4 °C for 1 min. Cell pellets were washed twice with cold PBS buffer and stored at −70 °C till use.

The P3P and P3Pe strains (Supplementary Table 1) were first cultured in 3 mL of M9 minimal media supplemented with 10 mM glycerol, 10 mM succinate, 10 mM acetate, 4 mM glycine, and 10 mM glycerate in 15 mL glass tubes. From these cultures, 1 mL of cells in the late exponential phase was collected and washed three times in M9 media. The washed cells were then used to inoculate 100 mL of M9 minimal media supplemented with 10 mM glycerol, 10 mM succinate, 10 mM acetate, and 4 mM glycine, with or without an additional 10 mM of glycerate, in 500 mL non-baffled shake flasks. These cultures were then incubated at 37 °C and 200 rpm in an Infors Minitron (Infors, Bottmingen, Switzerland). Cells from two biological replicates, along with their respective technical replicates, were harvested in mid-log phase (OD$_{600}$ = 0.3–0.5) and washed twice with phosphate buffer (12 mM phosphate buffer, 2.7 mM KCl, 137 mM NaCl, pH 7.4). The cell pellets, corresponding to an OD$_{600}$ of 3, were flash-frozen in liquid nitrogen and stored at −70 °C until further use.

For cell lysis, the cells were incubated at 90 °C for 15 min in 2% sodium lauroyl sarcosinate (SLS), followed by sonication for 15 s using a Vial Tweeter (Hielscher). Soluble proteins were reduced by incubation with 5 mM Tris(2-carboxy-ethyl) phosphine (TCEP) at 90 °C for 15 min and then alkylated with 10 mM iodoacetamide for 15 min at

25 °C. Protein concentrations were determined using a BCA protein assay kit (Thermo Fisher Scientific).

50 μg of protein were then digested with 1 μg of trypsin (Promega) in 0.25% SLS overnight at 30 °C. For SLS removal, trifluoroacetic acid (TFA) was added to a final concentration of 1.5%, incubated at room temperature for 10 min, and centrifuged at full speed on a tabletop centrifuge to pellet the precipitated detergent. The supernatant was purified using C18 Chromabond Microspin Columns (Macherey–Nagel). Cartridges were prepared by adding acetonitrile (ACN), followed by equilibration with 0.1% TFA. Peptides were loaded on equilibrated cartridges, washed with 5% ACN and 0.1% TFA-containing buffer, and finally eluted with 50% ACN and 0.1% TFA. Peptides were dried in a SpeedVac and reconstituted in 0.1% TFA before liquid-chromatography-mass spectrometry (LC-MS) analysis.

LC-MS analysis was carried out on an Exploris 480 instrument connected to an Ultimate 3000 RSLC nano and a nanospray flex ion source (all Thermo Scientific). Peptide separation was performed on a reverse phase HPLC column (75 μm × 42 cm) packed in-house with C18 resin (2.4 μm; Dr. Maisch). The following separating gradient was used: 94% solvent A (0.15% formic acid) and 6% solvent B (99.85% acetonitrile, 0.15% formic acid) to 25% solvent B over 40 min, and an additional increase of solvent B to 35% for 20 min at a flow rate of 300 nL/min. Peptides were ionized at a spray voltage of 2.3 kV, 445.12003 *m/z* was used as an internal calibrant. The funnel RF level was at 40, and heated capillary temperature was at 275 °C.

MS raw data was acquired on an Exploris 480 (Thermo Scientific) in data-independent acquisition mode (DIA). For DIA experiments full MS resolutions were set to 120.000 at *m/z* 200 and the full MS, AGC (Automatic Gain Control) target was 300% with an IT of 50 ms. AGC target value for fragment spectra was set at 3000%. 45 windows of 14 Da were used with an overlap of 1 Da from *m/z* 349.5–980.5. Resolution was set to 15,000 and IT to 22 ms. Stepped HCD collision energy of 25, 27.5, and 30% was used. MS1 data was acquired in profile, and MS2 DIA data in centroid mode.

Analysis of DIA data was performed using DIA-NN version 1.8[71] using a UniProt protein database[72] from *E.coli* MG1655 including common protein contaminants. For pyridoxine auxotrophic strains, C-terminal sequences of ThrA and Sad from their mutations, G472 and Q262, respectively, were removed to avoid peptide assignment limitations. The full tryptic digest was allowed with two missed cleavage sites, and oxidized methionines and carbamidomethylated cysteines. Match between runs and remove likely interferences were enabled. The neural network classifier was set to the single-pass mode, and protein inference was based on genes. The quantification strategy was set to any LC (high accuracy). Cross-run normalization was set to RT-dependent. Library generation was set to smart profiling. Nr. missed cleavages were set to 2, oxidized methionines were enabled as a variable, and cysteine carbamidomethylation as a fixed modification. The precursor false discovery rate (FDR) was set to 1%. DIA-NN recommended MS1 tolerance was internally set between 3.3 and 5.6 ppm. DIA-NN outputs were further evaluated using the SafeQuant script[73,74] modified to process DIA-NN outputs.

### Reporting summary

Further information on research design is available in the Nature Portfolio Reporting Summary linked to this article.

## Data availability

Raw reads of sequencing data are deposited at NCBI and can be accessed under BioProject PRJNA1002757 and PRJNA1010158. SIJ488 genome can be accessed under CP132594. X-ray structures can be accessed under PDB accession codes 8QMQ, 8QMR, 8QMS, and 8QMT. Proteomic data are available via ProteomeXchange with the identifier PXD048035. Source data are provided with this paper.

## Code availability

Python scripts for growth curve calculations and plots are available at Github [https://github.com/he-hai/growth2fig]. Python scripts for enzyme kinetics calculation are also available at Github [https://github.com/he-hai/enzymatic-tools].

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

## Acknowledgements

This study was funded by the Max Planck Society. P.A.G.-C. and T.J.E were also funded by EU Horizon 2020 Grant 862087 (Gain4Crops). We thank Maria Kowald and Pascal Pfister for their experimental assistance. We thank Beau Dronsella, Haozhe Chen, and Van Tuan Trinh for the scientific discussions. We thank the Research Core Facility Screening and Automation Technologies (SAT) of the Center for Synthetic Microbiology SYNMIKRO for Illumina sequencing. The X-ray diffraction data was collected at the P13 and P14 beamlines operated by EMBL Hamburg at the PETRA III storage ring (DESY, Hamburg, Germany), as well as at

ID30A-3 (MASSIF-3) of the ESRF (European Synchrotron Radiation Facility, Grenoble, France).

## Author contributions

Conceptualization, H.H. and T.J.E.; Methodology, H.H., P.A.G.-C., J.Z., S.B., D.S., N.P., and T.G.; Software, H.H. and T.G.; Investigation, H.H., P.A.G.-C., J.Z., S.B., J.K., P.C., M.Klein, M.Klose, and D.S.; Validation, H.H., P.A.G.-C., J.Z., N.P., and T.G.; Formal analysis, H.H., P.A.G.-G., J.Z., V.C-L., N.P., T.G., and T.J.E.; Resources, D.S., N.P., T.G., and T.J.E.; Writing – Original draft, H.H., P.A.G.-G., J.Z., N.P., and T.G.; Writing – review & editing, H.H., P.A.G.-C., J.Z., S.B., D.S., N.P., T.G., and T.J.E.; Visualization, H.H. and J.Z.; Funding acquisition, T.J.E.

## Funding

## Competing interests

The authors declare no competing interest.
