## [Peer Review File · Nature Communications]

Adaptive laboratory evolution recruits the promiscuity of succinate semialdehyde dehydrogenase to repair different metabolic deficiencies

Corresponding Author: Professor Tobias Erb

Version 0:

Reviewer comments:

Reviewer #1

(Remarks to the Author)

With great interest I was reading the manuscript by He and coworkers about the replacement of the GapA and Epd by the promiscuous succinate semialdehyde dehydrogenase Sad in the Gram-negative model bacterium *Escherichia coli*. The manuscript illustrates how genetic lesions can be relieved by a few genetic changes that unmask the promiscuous activity of a metabolic enzyme that is seemingly well understood. It also shows that underground metabolism needs to be investigated in a more systematic manner.

By employing a series of genetic and biochemical approaches He and coworkers demonstrated that mutations in the sad gene and the amplification of genomic segment containing the sad gene suppresses the growth defect of *E. coli* mutants with disrupted pathway required for sugar degradation and vitamin B6 biosynthesis.

I have a few suggestions that should be addressed by the authors to improve the manuscript.

Major comments

1. The observation that mutations within a gene encoding an enzyme with a low activity against a certain substrate can improve enzyme activity is quite common (see PMID: 23087246). The key publication by the lab of Dan Andersson showing real-time evolution of an enzyme having to low level activities should be cited.
2. It has also been described by various groups that selective gene amplification is a common mechanism to suppress the growth defect of a mutant or of a maladapted bacterial strain: PMIDs: 19686082, 23841635, 28294562, 29027347, 29915086, 30957856. Therefore, with respect to the identified mutations, the study by He and coworkers is not novel.
3. Supplementary Figures 2b and 2c: Why did the authors add casamino acids to the agar plates and not to the liquid medium? The difference in the medium composition can affect the outcome of the evolution experiment.
4. Page 5, line 92: The epd gapA double mutant was used to isolate suppressor mutants that could grow in minimal medium. Why did the authors start with two identical strains (EG1 and EG2)? This is quite uncommon because one could have sequenced the genome of the parental strain prior to the evolution experiment. In the later experiments, only one parental strain was used for ALE.
5. Page 6, line 116: Was the parental strain also sequenced? Are there deviations in the sequences of the reference genome SIJ488 and of the parental strain? What about the garP mutation in the evolved strains (Supplementary Table 2)? Is this sequence alteration present in the non-evolved parental strain?
6. Figure 6b: This experiment must be repeated, and the strain must be analyzed. In Supplementary Figure 2 it is shown that suppressor mutants appear after 80 h of incubation. Now, the authors state that the long lag phase of strain P3Pe sad is due to low expression of sad. It does not take 80 h to transcribe a gene in *E. coli*.

Minor comments

Figure 1: Also, a block in a metabolic pathway can redirect the metabolic flux to a promiscuous enzyme.

Supplementary Table 2 and Supplementary Figure 2 (also page 6, line 114): Did the six fastest growing suppressor mutants evolve in liquid medium or on agar plates? This is unclear.

Figures 1b and 2a: I recommend enlarging these figures because some parts are very small. In case the figure sizes are further reduced in the final version of the manuscript, they will be difficult to understand.

Page 4, line 35: A reference on vitamin B6 is missing here: 17822383, 27890703.

Page 5, lines 106 – 107: Why is growth variation a sign of adaptive evolution? It is simply the result of genetic instability of the strain due to spontaneous mutations.

Page 6, line 120: Why “putative amplification”? Gene amplification can be determined by Southern blotting. It would be interesting to know the frequencies of the different mutations, a question that can be answered by sequencing more suppressor mutants.

Page 7, lines 136 - 137: It is only a two-fold decrease in enzyme activity.

Page 9, lines 188 – 190: Instead of doing proteome analysis, overexpression of wild type and evolved sad gene (which is later shown) would have immediately provided an answer to this question.

Page 12, line 231: Only gabD needs to be overexpressed. Wild type sad allows already growth in the absence of the inducer. Please rephrase.

Page 14, line 278 (and throughout the manuscript): pyridoxine can be abbreviated (PN, see PLP).

Page 14, lines 296 – 297: better “E97A amino acid exchange”.

Page 15, line 300: Is the effector of YneJ known? If yes, please provide the correct reference.

Reviewer #2

(Remarks to the Author)

Summary

The authors present a relatively simple story detailing the molecular mechanisms by which *E. coli* can recruit succinate semialdehyde dehydrogenases to compensate for the loss of two different functions: the ability to synthesize pyridoxal 5'-phosphate, and the role of glyceraldehyde 3-phosphate dehydrogenase in glycolysis. They use a slew of genetic and biochemical characterizations to show the specific mechanisms.

Overall I found the manuscript clearly written, convincing, and detailed. I have only a few minor comments.

Comments

The ALE on plates needs additional details on the timing and approximate colony number (if they were visible) - the methods give only “Colonies were scraped and re-streaked to Medium X plates three times.”

L.120: “amplified genomic regions that encoded sad” - please supply the estimated copy number of the amplification. Also “A putative amplification of sad was also found in EG1.2x” - why is this considered “putative”?

L.122: “EG2.1t had accumulated a frameshift in thrA shortening the coding protein.” The frameshift is 87 bp from the end of the protein, or ~29 codons, while ~1/21 codons are stop codons. Does the frameshift indeed result in a premature stop or just a change in the downstream AA sequence?

L.126: “we aimed at elucidating the mutation(s) of the sad gene” → “we aimed at elucidating the effects of the mutation(s) of the sad gene” or the “functional consequences of”, or similar.

Fig. 4f: It would be informative to know the approximate expression fold-increase under 50uM IPTG

L.296: “(rather unspecific)” → “most likely irrelevant” or similar, or is the proposal that these are somehow advantageous from a general ALE perspective?

L. 344: “was that discovery” → “was the discovery”

Supp. material MG1655 and SIJ488 genome comparison - the DOI for the differences gives an error - is the DOI correct?

Clarify Supplementary Figure 3 - if this is an amplification, why do the surrounding regions have a coverage of 0? It would also be informative to see the inferred genome structure in this region, for example to understand whether promoter capture is responsible for some increase in expression.

Version 1:

Reviewer comments:

Reviewer #1

(Remarks to the Author)

I am happy with the revision of the manuscript, which was significantly improved.

Reviewer #2

(Remarks to the Author)

The authors have satisfactorily addressed my concerns.

Reviewer #1 (Remarks to the Author):

With great interest I was reading the manuscript by He and coworkers about the replacement of the GapA and Epd by the promiscuous succinate semialdehyde dehydrogenase Sad in the Gram-negative model bacterium Escherichia coli. The manuscript illustrates how genetic lesions can be relieved by a few genetic changes that unmask the promiscuous activity of a metabolic enzyme that is seemingly well understood. It also shows that underground metabolism needs to be investigated in a more systematic manner.

By employing a series of genetic and biochemical approaches He and coworkers demonstrated that mutations in the sad gene and the amplification of genomic segment containing the sad gene suppresses the growth defect of E. coli mutants with disrupted pathway required for sugar degradation and vitamin B6 biosynthesis.

I have a few suggestions that should be addressed by the authors to improve the manuscript.

We thank the reviewer for the positive evaluation of our work.

Major comments

1. The observation that mutations within a gene encoding an enzyme with a low activity against a certain substrate can improve enzyme activity is quite common (see PMID: 23087246). The key publication by the lab of Dan Andersson showing real-time evolution of an enzyme having to low level activities should be cited.

We thank the reviewer for this suggestion and cite the literature now in context (ref 4).

2. It has also described by various groups that selective gene amplification is a common mechanism to suppress the growth defect of a mutant or of a maladapted bacterial strain: PMIDs: 19686082, 23841635, 28294562, 29027347, 29915086, 30957856. Therefore, with respect to the identified mutations, the study by He and coworkers is not novel.

We appreciate your insightful comment. Indeed, some of the suppressor mechanisms are similar to what has been reported (i.e., gene duplication or active site mutation, which are commonly observed in ALE). However, one core message our study conveys is the intrinsic plasticity of an **individual** enzyme as reservoir for evolutionary innovation. We show in detail how evolution recruits the promiscuous activity of a **single** enzyme (succinate semialdehyde dehydrogenase, SSADH) to suppress **multiple** metabolic deficiencies. Our findings condense **different** evolutionary mechanisms applied into one **single** functional unit and showcase the potential of an “ordinary” detoxification enzyme as **evolutionary innovation hotspot**. In our revised manuscript, we hope to have worked out the novelty of our findings better.

3. Supplementary Figures 2b and 2c: Why did the authors add casamino acids to the agar plates and not to the liquid medium? The difference in the medium composition can affect the outcome of the evolution experiment.

Thank you for asking. Indeed, we used different media. Especially for solid medium, we added more nutrients to increase the chance for foundational suppressor mutations. To highlight this fact, we revised the text as follows:

“... additionally conducted an adaptive laboratory evolution (ALE) experiment on solid medium (supplemented with casamino acids, tryptophan and thiamine) to potentially increase the spectrum of mutations”

We have also revised the Supplementary Figure 2 accordingly.

4. Page 5, line 92: The *epd gapA* double mutant was used to isolate suppressor mutants that could grow in minimal medium. Why did the authors start with two identical strains (EG1 and EG2)? This is quite uncommon because one could have sequenced the genome of the parental strain prior to the evolution experiment. In the later experiments, only one parental strain was used for ALE.

Thank you for asking. The two strains are different isolates from the same knock-out step. We intended to increase the likelihood of suppressor mutants and potentially expand the spectrum of mutations, all without the need of prior costly and time consuming parental strain sequencing. To achieve this, we employed different strategies, replicating parental strain in the first case and replicating culture in the later. In our experience, this strategy helps us to pinpoint relevant mutations better, as the parallel evolving strains accumulate less “piggy-back” mutations. To clarify the strain origin, we revised the text as follows:

“We picked two independent colonies from the same knock-out step, designated as biological replicates, and labeled as EG1 and EG2”

5. Page 6, line 116: Was the parental strain also sequenced? Are there deviations in the sequences of the reference genome SIJ488 and of the parental strain? What about the *garP* mutation in the evolved strains (Supplementary Table 2)? Is this sequence alteration present in the non-evolved parental strain?

Yes, we sequenced both parental strains EG1 and EG2. As mentioned by you, a T172I mutation in *garP* was accidentally introduced to both parental strains during strain construction, shown in Supplementary Table 2. Since this mutation was already present in the parental strains and appears to be irrelevant to pyridoxine metabolism, we did not discuss it in the main text. However, in response to your comment, we now revised the title of the Supplementary Table 2 to:

“*Breseq* identified sequence deviations of the parental (EG1 and EG2) and evolved $\Delta epd \Delta gapA$ strains from the SIJ488 reference genome.”

And added a footnote to this table, “Note that a T172I mutation in *garP* was inadvertently introduced to both parental strains during strain construction.”

6. Figure 6b: This experiment must be repeated, and the strain must be analyzed. In Supplementary Figure 2 it shown that suppressor mutants appear after 80 h of incubation. Now, the authors state that the long lag phase of strain P3Pe *sad* is due to low expression of *sad*. It does not take 80 h to transcribe a gene in *E. coli*.

The reviewer is correct that it does not require 80 hours to transcribe and translate a gene in *E. coli* (even at every low growth rates). We apologize if we explained the “lag phase” phenomenon insufficiently. We wanted to point out that in these strains, it is about reaching a threshold of intracellular enzyme (i.e., Sad) levels to carry sufficient biosynthetic flux to enable growth. Please note that already for PLP biosynthesis in the EG strain (<0.01%

biomass), “normal” expression level of Sad did not support sufficient flux, requiring mutations improving catalytic efficiency or enzyme levels. The P3P strain on the other hand requires even higher flux for the biosynthesis of serine and PEP (~7% biomass), so it also requires much higher steady-state intracellular Sad levels.

To support this point, we added new data to Fig 6b and provided a new Supplementary Fig 15, in which we tested growth at different IPTG concentrations, when expressing Sad from a plasmid. In the updated Fig 6b and the new Supplementary Fig 15, we show that the strain (P3Pe Δsad with Sad expression plasmid) has no lag phase at 5 mM IPTG induction. In contrast, the strain shows a long lag phase at low IPTG concentrations (1 mM), and even no growth in the absence of IPTG, respectively. Additionally, we note that we had to use IPTG in our preculture, as no IPTG in our preliminary experiment resulted in no growth of this strain. While the P3Pe strain seemed to be consistently upregulated (Fig 6c and Supplementary Fig 14).

We revised the main text as follows:

“Overexpressing *sad* from a plasmid in strain P3Pe Δsad under high concentrations of IPTG rescued growth (light blue line in Fig. 6b and Supplementary Figure 15)”

“Indeed, proteome analysis showed that in P3Pe, the *sad-glsB-yneG* operon seemed to be consistently upregulated, the levels of all proteins were increased by one and two orders of magnitude compared to the parent strain P3P”

We also added a note in the legend of Figure 6b, “50 μ M IPTG was used in the preculture.”

Minor comments

Figure 1: Also, a block in a metabolic pathway can redirect the metabolic flux to a promiscuous enzyme.

We thank the reviewer for the valuable input. We have added a new panel (a) in Figure 1 accordingly, and revised the relevant text and figure contents.

Supplementary Table 2 and Supplementary Figure 2 (also page 6, line 114): Did the six fastest growing suppressor mutants evolve in liquid medium or on agar plates? This is unclear.

We apologize for the lack of clarity. We added the information to Supplementary Table 1 and the legend of Supplementary Figure 2. Specifically, for the six fastest growing suppressor strains, EG1.2x, EG2.1t, and EG2.2s evolved in liquid medium, while EG2.3a, EG2.4a, as well as EG2.5a, evolved on agar plates.

Figures 1b and 2a: I recommend enlarging these figures because some parts are very small. In case the figure sizes are further reduced in the final version of the manuscript, they will be difficult to understand.

Thank you! We enlarged the figures and hope to still adhering to the requirements of the journal.

Page 4, line 85: A reference on vitamin B6 is missing here: 17822383, 27890703.

We thank the reviewer for this remark. We have added the citations.

Page 5, lines 106 – 107: Why is growth variation a sign of adaptive evolution? It is simply the result of genetic instability of the strain due to spontaneous mutations.

Indeed, mutations occur spontaneously and only allow for growth if beneficial (i.e., restoring pyridoxine biosynthesis in strains EG1 & EG2). Since, the nature and timing of these mutations strongly vary, this variability leads to differences among replicate cultures. However, as shown in Supplementary Figure 2a, under relaxing condition (i.e., 120 nM pyridoxine), the strains grew identically across replicates.

We revised the sentence in the main text for clarity:

“... growth of individual replicate cultures could be observed after prolonged incubation”

Page 6, line 120: Why “putative amplification”? Gene amplification can be determined by Southern blotting. It would be interesting to know the frequencies of the different mutations, a question that can be answered by sequencing more suppressor mutants.

We used the term of “putative amplification” because they were predicted from short-read sequencing data, not from long-read sequencing data, which would – in principle – more directly show multiple copies in a single read (see below). However, we were not able to use long-read sequencing techniques (Nanopore or PacBio) to directly show the amplification, because the region is too large to be covered by a single read (>100 kb, see the newly added Supplementary Table 3).

Yet, our prediction has high confidence: We used *breseq* pipeline to map reads (75 bp) to the reference genome, which predicted “new junctions” at breakpoints (ref 28 and the “New Junction” section of the *breseq* manual <https://barricklab.org/twiki/pub/Lab/ToolsBacterialGenomeResequencing/documentation/output.html#evidence-display>). From the type of the “new junction” (chimeric reads from both ends of the amplified region) and the increased coverage of the amplified region, the “new junction” prediction was manually annotated with high confidence as “amplification” (tandem repeat). To avoid confusion, we revised the term to “amplification” in the text and Supplementary Table 2.

In the new Supplementary Table 3, we show the calculated number of repeats through two independent ways: (a) coverage fold-change of the amplified region to the surrounding region; (b) frequency of chimeric reads to total reads that mapped to the breakpoints (for example, frequency of 50% would indicate a duplication). Both calculations predict copy numbers of approximately 30 in EG1.2x and 80 of EG2.3a, EG2.4a and EG2.5a. Because migration rates or band intensity comparisons from Southern blots would not give better resolution, we decided not include such experiments.

We changed the Supplementary Fig 3 to plot “Normalized coverage” and revised its legend,

“*breseq* predicted “New Junction” in EG2.3a, EG2.4a, EG2.5a, and EG1.2x strains (Supplementary Table 1 and 2) at the points marked with the pink lines. From the type of “New Junction” and increased coverage, we manually annotated the regions in between the pink lines “gene amplification” (tandem repeat). The line plots show the

normalized coverage from *breseq* mapping. Approximately 80× and 30× increases of coverage of the amplified regions were estimated. Coverage statistics are detailed in Supplementary Table 3.”

Page 7, lines 136 - 137: It is only a two-fold decrease in enzyme activity.

Thank you, we thus revised the sentence:

“the Q262R variant showed a significantly decreased activity ...”

Page 9, lines 188 – 190: Instead of doing proteome analysis, overexpression of wild type and evolved *sad* gene (which is later shown) would have immediately provided an answer to this question.

The reviewer is right that overexpression of the *sad* gene could mimic the growth phenotype, as we also showed later in our complementation experiment. However, proteomics was part of our routine OMICS workflow and provided us with direct evidence of protein overproduction within the evolved strains.

We revised the text:

“... *sad* (and surrounding genomic regions) had been predicted to be amplified during ALE”

Page 12, line 231: Only *gabD* needs to be overexpressed. Wild type *sad* allows already growth in the absence of the inducer. Please rephrase.

Thank you, we revised to “fully restored”:

“Sad Q262R fully restored growth of the *pydxnA* strain in the absence of pyridoxine even without IPTG induction, while Sad WT and GabD both required induction”

Page 14, line 278 (and throughout the manuscript): pyridoxine can be abbreviated (PN, see PLP).

We now use PN in the main text.

Page 14, lines 296 – 297: better “E97A amino acid exchange”.

Corrected.

Page 15, line 300: Is the effector of YneJ known? If yes, please provide the correct reference.

GABA and glutamine were reported to be effectors of YneJ (ref 44 & 45), however the specificity of the effector binding domain of YneJ was not investigated yet. The effector binding domain of YneJ is similar to other LysR type TF family members (ref 46), that are known to bind to bind diverse effectors, including sugar phosphates, amino acids, organic acids in glycolysis (newly add ref 47 and 48). We therefore revised the text with new citations:

“Notably, YneJ has been recently shown to regulate the expression of the *sad* operon (*sad-glsB-yneG*) (see genome context in Supplementary Figure 3a), which is induced by glutamine. Because E97 is located in the

effector binding domain of YneJ (Supplementary Figure 13a) and LysR-type transcription factors are known to bind diverse effectors, including sugar phosphates and amino acids^{47,48}, we speculated that ...”

Reviewer #2 (Remarks to the Author):

Summary

The authors present a relatively simple story detailing the molecular mechanisms by which *E. coli* can recruit succinate semialdehyde dehydrogenases to compensate for the loss of two different functions: the ability to synthesize pyridoxal 5'-phosphate, and the role of glyceraldehyde 3-phosphate dehydrogenase in glycolysis. They use a slew of genetic and biochemical characterizations to show the specific mechanisms. Overall I found the manuscript clearly written, convincing, and detailed. I have only a few minor comments.

We thank the reviewer for the support and helpful comments.

Comments

The ALE on plates needs additional details on the timing and approximate colony number (if they were visible) - the methods give only “Colonies were scraped and re-streaked to Medium X plates three times.”

We apologize for the lack of details. The strains were scraped using inoculation loops when they were fully grown after ~ 3 days, and directly re-streaked to fresh plates. This procedure was repeated three times. Then single colonies were picked up. Using this technique, we were not able to count colony numbers.

We revised the method section as below:

“Cell materials were scraped with inoculation loops when fully grown (around 3 days) and directly re-streaked to fresh Medium X plates. This procedure was done three times (Supplementary Figure 2).”

L.120: “amplified genomic regions that encoded sad” - please supply the estimated copy number of the amplification. Also “A putative amplification of sad was also found in EG1.2x” - why is this considered “putative”?

The copy number were estimated to be 80 for EG2.3a, EG2.4a, as well as EG2.5a and 30 for EG1.2x. We replotted the coverage data in Supplementary Fig 3a to “normalized coverage” directly indicating copy numbers. We also added a new Supplementary Table 3 for the copy number estimation.

In the new Supplementary Table 3, we show the calculated number of repeats through two independent ways: (a) coverage fold-change of the amplified region to the surrounding region; (b) frequency of chimeric reads to total reads that mapped to the breakpoints (for example, frequency of 50% would indicate a duplication). Both calculations predict copy numbers of approximately 30 in EG1.2x and 80 of EG2.3a, EG2.4a and EG2.5a.

The main text was also revised:

“... while EG2.3a, EG2.4a, and EG2.5a were predicted by the *breseq* pipeline to carry amplified genomic regions that encoded *sad* (~80×, Supplementary Figure 3a and Supplementary Table 3). An amplification of *sad* was also found in EG1.2x (~30×, Supplementary Figure 3a and Supplementary Table 3) ...”

Regarding the “putative amplification”, as the response to Reviewer 1, we used the term of “putative amplification” because they were predicted from short-read sequencing data, not from long-read sequencing data, which would – in principle – more directly show multiple copies in a single read. However, our prediction has high confidence (explained below). To avoid confusion, we revised the term to “amplification” in the text and Supplementary Table 2.

We used *breseq* pipeline to map the reads (75 bp) to the reference genome, it predicted “new junctions” at breakpoints (ref 28 and the “New Junction” section of the *breseq* manual <https://barricklab.org/twiki/pub/Lab/ToolsBacterialGenomeResequencing/documentation/output.html#evidence-display>). From the type of the “new junction” (chimeric reads from both ends of the amplified region) and the increased coverage of the amplified region, the “new junction” prediction was manually annotated with high confidence as “amplification” (tandem repeat).

We were not able to use long-read sequencing techniques (NanoPore or PacBio) to directly show the amplification, because the region after amplification was estimated to be too large (> 140 kb, see the newly added Supplementary Table 3) to be covered by a single read.

L.122: “EG2.1t had accumulated a frameshift in *thrA* shortening the coding protein.” The frameshift is 87 bp from the end of the protein, or ~29 codons, while ~1/21 codons are stop codons. Does the frameshift indeed result in a premature stop or just a change in the downstream AA sequence?

The frameshift changed the reading frame to +2, resulting in a change in the downstream AA sequence, which indeed affected both AA sequence and length of the protein. We added a figure to Supplementary Figure 3b and revised the text:

“Finally, EG2.1t had accumulated a frameshift in *thrA*, changing downstream amino acid sequence and shortening it by three amino acids (Supplementary Figure 3b).”

L.126: “we aimed at elucidating the mutation(s) of the *sad* gene” → “we aimed at elucidating the effects of the mutation(s) of the *sad* gene” or the “functional consequences of”, or similar.

We corrected the sentence.

Fig. 4f: It would be informative to know the approximate expression fold-increase under 50uM IPTG

Using data from Schuster & Reisch 2022 (DOI: 10.1128/aem.00939-22, ref 37), the expression level at this concentration is estimated to be increased ~200-fold. The pDLIXX plasmids used in Schuster & Reisch study has the same origin or replication, CloDF13, and the same promoter, P_{LacO-1}, as our expression plasmids

(Supplementary Table 5). The Schuster & Reisch report showed leaky expression in the absence of IPTG, and similar expression level between 16 - 400 μ M IPTG. This seems very comparable with our observation that strains grew similarly at 50 μ M and 200 μ M IPTG. Therefore, we modified the text in Line 238-240:

“Sad Q262R fully restored growth of the pydxnA strain in the absence of PN even without IPTG induction, while Sad WT and GabD both required induction (Fig. 4e and f), at IPTG levels resulting ~200-fold increased protein level compared to the non-induced state.”

L.296: “(rather unspecific)” → “most likely irrelevant” or similar, or is the proposal that these are somehow advantageous from a general ALE perspective?

We corrected the sentence as suggested.

L. 344: “was that discovery” → “was the discovery”

Corrected.

Supp. material MG1655 and SIJ488 genome comparison - the DOI for the differences gives an error - is the DOI correct?

We apologize for this error. We fixed the DOI and made sure that the file is now accessible.

Clarify Supplementary Figure 3 - if this is an amplification, why do the surrounding regions have a coverage of 0? It would also be informative to see the inferred genome structure in this region, for example to understand whether promoter capture is responsible for some increase in expression.

We apologize for the lack of clarity. The coverage of surrounding regions was not 0, but too low when shown at the same scale as the amplified region. We revised the figure to “Normalized coverage” and added Supplementary Table 3 to show the statistics.

The breakpoints of the amplified region were indicated in the figure with pink lines. The promoter of *sad* gene is inside the amplified region.

Reviewer #2 (Remarks on code availability):

The code is available, with the figure code having a readme and a demo with some details on how to run. The test data runs after cloning the repo. The docs of the enzymatic tools code is less extensive.

We added a section in the readme to briefly explain. There are also commented lines inside the script to help users.